# Far-field coherent thermal emission from polaritonic resonance in individual anisotropic nanoribbons

Sunmi Shin[1], Mahmoud Elzouka[2], Ravi Prasher[2,3] & Renkun Chen [1,4]

Coherent thermal emission deviates from the Planckian blackbody emission with a narrow spectrum and strong directionality. While far-field thermal emission from polaritonic resonance has shown the deviation through modelling and optical characterizations, an approach to achieve and directly measure dominant coherent thermal emission has not materialised. By exploiting the large disparity in the skin depth and wavelength of surface phonon polaritons, we design anisotropic $SiO_2$ nanoribbons to enable independent control of the incoherent and coherent behaviours, which exhibit over 8.5-fold enhancement in the emissivity compared with the thin-film limit. Importantly, this enhancement is attributed to the coherent polaritonic resonant effect, hence, was found to be stronger at lower temperature. A thermometry platform is devised to extract, for the first time, the thermal emissivity from such dielectric nanoemitters with nanowatt-level emitting power. The result provides new insight into the realisation of spatial and spectral distribution control for far-field thermal emission.

[1] Materials Science and Engineering Program, University of California, San Diego, CA 92093, USA. [2] Energy Storage and Distributed Resources Division, Lawrence Berkeley National Laboratory, Berkeley, CA 94720, USA. [3] Department of Mechanical Engineering, University of California, Berkeley, CA 94720, USA. [4] Department of Mechanical and Aerospace Engineering, University of California, San Diego, CA 92093, USA. Correspondence and requests for materials should be addressed to R.P. (email: rsprasher@lbl.gov) or to R.C. (email: rkchen@ucsd.edu)

Thermal radiation, as described by Planck's law, is generally considered incandescent in classic textbooks. However, as Planck himself originally noted[1], the emitting energy distribution, including the broadband nature in both spectral and spatial domains, no longer holds if the characteristic size is smaller than the thermal wavelength ($\lambda_T \sim 10\ \mu m$ at room temperature). Recent rapid development in near-field radiative heat transfer[2–6] has revealed extraordinary behaviours when the distance between the emitter and receiver is smaller than $\lambda_T$, including giant enhancement in the heat flux and a coherent feature resulting from the tunnelling of evanescent waves by surface phonon polaritons (SPhPs) or surface plasmon polaritons (SPPs). Although coherence has been extensively investigated in the near field via evanescent waves, observation of coherence in far-field is still significantly limited. Far-field emission from an emitter with a size smaller than $\lambda_T$ can result in polarised thermal emission that could be relevant for broader applications for which the near-field gap is inaccessible[7–13]. Recent work has shown the feasibility of spectral and spatial control of far-field thermal emission from engineered microstructures. For instance, Greffet et al. demonstrated coherent emission, featured with a narrow spectral and angular range, achieved by the grating of a polaritonic material (i.e., SiC) with a grating period comparable to $\lambda_T$ to couple to SPhPs[14]. More recently, through fluctuational electrodynamics modelling, Fernández-Hurtado et al.[15] showed highly directional emissivity from anisotropic polaritonic structures, which was found to originate from the guided modes whose dispersion relation becomes highly dispersive around the Reststrahlen band. This feature resulted in super-Planckian thermal radiation between two anisotropic objects when they are oriented along the dominant emission direction, which was also experimentally demonstrated by Thompson et al.[16].

These earlier findings provide important insights into the design of an emitter with coherent far-field thermal emission. To achieve dominantly coherent thermal emission in the far field, it is desirable to quench the incoherent emission. Bulk polar dielectric materials, such as $SiO_2$, generally behave like a blackbody in the broad infrared (IR) range, except within the Reststrahlen band[17] (negative permittivity) that shows high reflectance (Fig. 1b). For $SiO_2$, the skin depth ($\delta$) at 10-$\mu m$ wavelength is on the order of 1 $\mu m$[18]. Therefore, one could utilise an emitter smaller than $\delta$ to suppress the broadband emission. This volumetric absorption regime was theoretically studied by Golyk et al.[18] for cylindrical $SiO_2$ and SiC nanorods, where the incoherent emission was greatly suppressed from the near-blackbody limit for rod diameters much smaller than $\delta$. However, the small rod diameter, now also much smaller than $\lambda_T$, would limit the resonant effect and diminish the potential coherent thermal emission. Due to this constraint, previously studied nanorods showing the unusual polarised thermal radiation behaviours would all have low overall emissivity[18–21], as modelled by Golyk et al.[18].

Here, we seek to design a nanoscale emitter that quenches the incoherent broadband emission without compromising the coherent resonance. Our strategy is to exploit another degree of anisotropy using rectangular polar dielectric nanoribbons, which have two characteristic length scales, the thickness and width, to separately match $\delta$ and $\lambda_T$, respectively. As shown in Fig. 1a, we utilised a much smaller thickness (100 nm) relative to $\delta$ to suppress the broadband incoherent thermal emission originated from the volumetric effect in cylindrical nanorods[18,19]. Moreover, we designed the width of the nanoribbons to be comparable to $\lambda_T$, which is expected to result in enhanced coherent thermal emission because of the resonance of the SPhPs. Guided modes by SPhPs shrink the wavelength ($\lambda_{SPhP}$) at the surface, as shown in Fig. 1c; furthermore, the finite cross-section determined by the width and thickness intensifies the absorption efficiency. A size comparable to integer multiples of the half wavelength maximises

the resonant enhancement. With this design, we can simultaneously enhance the coherent emission and quench the background broadband emission, thereby achieving dominantly coherent thermal emission within the narrow Reststrahlen band.

A perceived experimental challenge to directly probing the thermal emission from these emitters is the exceedingly low emitting power (down to nW, assuming 10-$\mu m$ width, 100-$\mu m$ length, 100-nm thickness, $\varepsilon = 0.2$, $T = 100$–400 K). To our knowledge, direct thermal emissivity ($\varepsilon$) measurements from dielectric nanoheaters have not yet been achieved because of the low emissive power. Prior $\varepsilon$ measurements have only been achieved on metallic nanoheaters, such as W, Pt, and carbon nanotubes[9,10,21,22], where the specimen also served as a joule heater and thermometer. Measurements on these metallic nanostructures revealed the important role of SPP resonance on emissivity. However, due to the high frequency of SPPs, their effect is only important at elevated temperature. Unfortunately, the dielectric itself cannot provide heating or thermometry, and any metallic addenda could impact the emission properties of the nanoscale specimen. Although recent scanning probe techniques[11,17,23] have enabled the detection of local near-field radiation intensity from nanoscale objects[24], these measurements are not suitable for far-field. Far-field optical measurements of thermal emission (e.g., Fourier-transform infrared spectroscopy) require a substrate and must be performed at high temperature to obtain detectable infrared signals[9,10,14], both of which can alter the emission behaviours. The substrate itself could introduce parasitic signals by refraction and reflection at the interface[25], and the thermalisation at high temperature could reduce the quality (Q) factor because of the reduced lifetime of optical phonons[26,27].

In this work, to directly measure the far-field thermal emissivity, $\varepsilon$, from single anisotropic nanoribbons of polar dielectrics with a perceived high degree of coherent thermal emission, we devise a novel thermometry platform by combining a fin model with sensitive thermometry. This platform allows us to observe over 8.5-fold enhancement in $\varepsilon$ of $SiO_2$ nanoribbons compared with that of thin-film $SiO_2$ of the same thickness. Importantly, through experimental observation and computational analysis, we are able to attribute this enhancement to the coherent resonant effect of SPhPs enabled by the anisotropic nanoribbon design. This resonance effect is further confirmed by the observed increasing emissivity at lower temperature, as the suppression of phonon broadening and increased phonon lifetime at low temperature lead to enhancement of the coherence and Q factor.

## Results

**Design of anisotropic nanoribbons.** We designed the anisotropic nanoribbons based on the relevant length scales of $\delta$[18] and $\lambda_{SPhP}$, as shown in Fig. 1a. The thickness of the ribbons was fixed at 100 nm to suppress the broadband emission. We estimated the volumetric emissivity to be 0.062 for this thickness at room temperature, which is much smaller than the bulk emissivity (~0.9). The design of the width of the ribbon was dependent on $\lambda_{SPhP}$. It should be noted that the wavelength of SPhPs at the $SiO_2$/vacuum interface is different from that in free space (~10 $\mu m$); it is dependent on the geometry of the $SiO_2$ ribbons because the dispersion of SPhPs is now controlled by the coupling between the optical phonon in $SiO_2$ and photons. We determined the dispersion of a thin film using the following analytical formula (see Supplementary Note 8):

$$\frac{\varepsilon_{SiO_2}}{\varepsilon_0} = -\frac{k_{t,SiO_2}}{k_{t,0}} \coth\left(\frac{t}{2i} k_{t,SiO_2}\right), \quad (1)$$

where $\varepsilon_0$ and $\varepsilon_{SiO_2}$ $(= \varepsilon' + i\varepsilon'')$ are the permittivity in vacuum and $SiO_2$[28], respectively; $k_{t,i}$ is the transverse vector

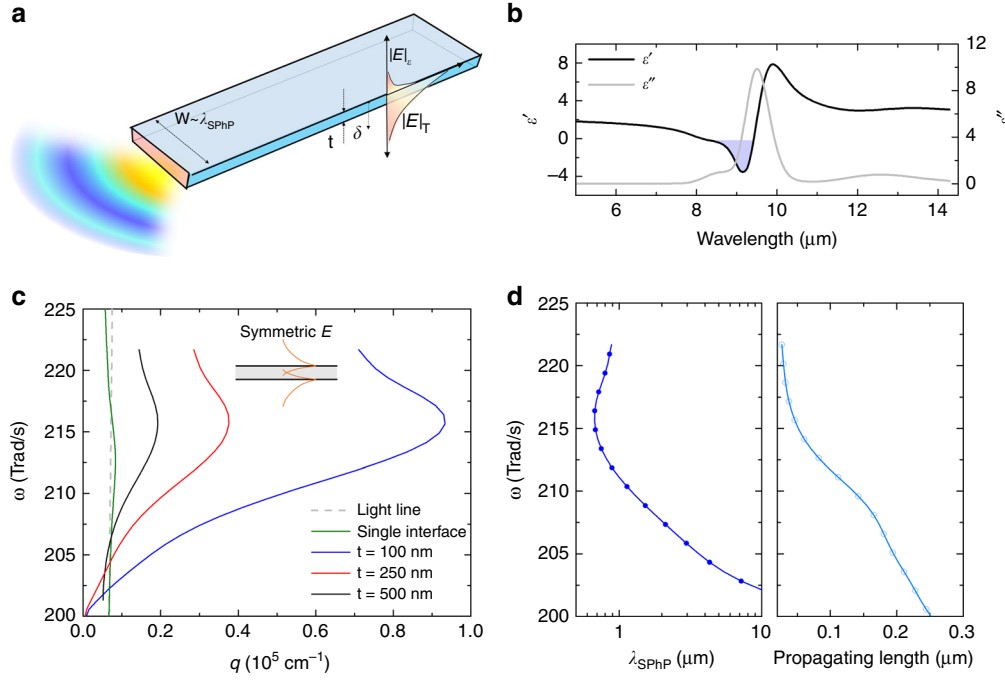

**Fig. 1** Anisotropic nanoribbon for localised resonance by polaritons. **a** Schematic illustration of a long nanoribbon with rectangular cross-section with thickness ($t$) and width ($W$). Here, $t$ is thinner than the skin depth ($\delta$), leading to significant transmitted intensity of the electromagnetic wave perpendicular to the ribbon ($|E|_T$) compared with the absorbed intensity ($|E|_\varepsilon$). Otherwise, the optical response in the parallel direction to the plane would yield higher emission because the cross-sectional length scale with the integer multiples of half wavelength supports the localised resonance modes, enhancing the absorption cross-section. **b** Plots of the real and imaginary permittivity of $SiO_2$. The region of the Reststrahlen band is coloured. **c** Dispersion relation of SPhP supported by thin films of different thickness, calculated from the approximate solution Eq. (1). The energy range is within the Reststrahlen band. **d** SPhP wavelength and propagating length for a 100 nm thick thin film, corresponding to the dispersion curve

$\left(k_{t,i}^2 + k_p^2 = \varepsilon_i k_0^2\right)$ in the medium $i$; $k_p$ is the propagating vector along the surface ($k_p = q + i\kappa$, where $q$ and $\kappa$ are the real and imaginary part of the momentum vector, respectively); and $k_0$ is the free space vector. Equation (1) is the solution for the even modes of two branches of the surface waves by electric fields at the top and bottom surfaces, and this case represents the dominant contributions for the efficient absorption by confining the energy from the free space into the small physical cross-section area[29]. The confinement becomes more significant with a smaller thickness, due to the stronger interaction of the two symmetric evanescent surface waves (Fig. 1c). The analytical equations are summarised in the Supplementary Information (see Supplementary Note 8) for the major available fundamental modes by different polarisations (namely, symmetric TM, asymmetric TM, and TE modes). The calculated dispersions, shown in Fig. 1c, indicate that the wavelength of SPhPs ranges from 0.8 to 1.8 μm, which is approximately 5–10 times smaller than the free-space wavelength, for a $SiO_2$ layer thickness of 100 nm. We designed the width (e.g., ~5 and 10 μm) to cover the integer multiples of the half wavelength of SPhP, to study the size effect on the resonant enhancement. The realised widths on the fabricated nanoribbons were 11.5 and 6.28 μm (Fig. 2).

**Direct thermal emissivity measurement platform**. We developed a microscale thermometry platform to measure the radiative heat loss of single $SiO_2$ nanoribbons, as shown in Fig. 2a–f (see Methods for the detailed fabrication process). We used a suspended device such that the emission comes entirely from the nanoribbon without any influence of a substrate. We also implemented a modulated AC heating technique (Fig. 3b) to

achieve sensitive thermometry with ~0.1 mK resolution[30–32]. The basic idea to extract the radiative heat loss was to utilise the thermal fin model by systematically varying the length of the ribbons. Considering heat conduction along the length ($L$) of a nanoribbon with consideration of the radiative heat loss characterised by the heat transfer coefficient $h$, where $h = 4\varepsilon\sigma T^3$ ($\varepsilon$ is the emissivity, $\sigma$ is the Stefan–Boltzmann constant, and $T$ is the temperature), the temperature rise along the ribbon can be described by the fin model:

$$\theta = \theta_h \cosh(mx) + \frac{\theta_s - \theta_h \cosh(mL)}{\sinh(mL)} \sinh(mx). \quad (2)$$

Here, $\theta$ ($\equiv T - T_0$) is the temperature rise from the ambient temperature ($T_0$) as a function of the $x$-coordinate along the length ($L$); $\theta_h$ and $\theta_s$ are the temperature rise at the hot and cold reservoirs, respectively; and $m$ is the fin parameter:

$$m = \sqrt{\frac{hP}{\kappa A_c}}. \quad (3)$$

Here, $\kappa$ is the intrinsic thermal conductivity of the ribbon, and $P$ and $A_c$ are the perimeter and cross-sectional area, respectively. The fin model notably allows us to quantitatively describe the combined energy transfer of light and heat in the form of radiation out of the surface and conduction through the volume. We summarise the temperature rise ratio of heating to the sensing

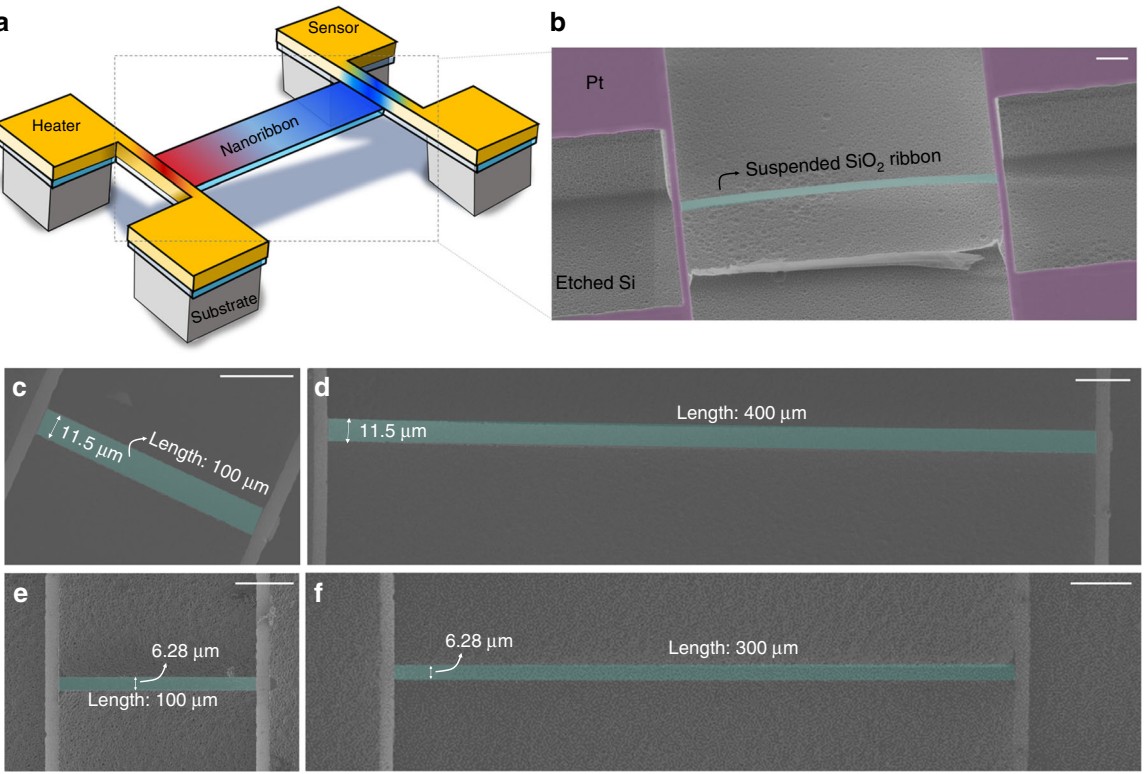

**Fig. 2** Suspended thermal transport measurement micro-device with monolithic $SiO_2$ nanoribbon. **a** Schematic illustration and **b** SEM image of the device showing the suspended heater and sensor electrodes (Pt) and a monolithic $SiO_2$ nanoribbon across the electrodes. **c–f** SEM images of $SiO_2$ nanoribbons of various sizes. All the scale bars represent 30 μm

side, $\gamma$, as

$$\frac{\theta_s}{\theta_h} = \frac{1}{\cosh(mL) + \frac{G_b}{\kappa A_c m}\sinh(mL)} \equiv \gamma. \tag{4}$$

In the absence of radiative heat loss, $m$ becomes 0; therefore, Eq. (4) can be simplified as

$$\left.\frac{\theta_s}{\theta_h}\right|_{m=0} = \frac{G_s}{G_b + G_s} \equiv \gamma_{\text{lossless}}, \tag{5}$$

where $G_s$ $(=\kappa A_c/L)$ is the thermal conductance of the sample and $G_b$ is the thermal conductance of metal beams (see Supplementary Note 4). Finally, we can define the ratio of $\gamma$ to $\gamma_{\text{lossless}}$ as $\Delta$ $(\gamma/\gamma_{\text{lossless}})$, where $\gamma_{\text{lossless}}$ is determined using short samples with bulk thermal conductivity. By comparing the $\Delta$ of samples with varying lengths, as shown in Figs. 2 and 3a, we can determine $m$ and eventually $h$ and $\varepsilon$.

The measured apparent thermal conductance includes the overall influences of both heat propagation through the solid volume and radiation out of the surface ($G_{\text{rad}}$). To apply this measurement technique to low-dimensional materials, high-resolution thermometry is needed to resolve the radiative loss from the net thermal conductance. It is especially important to be able to detect the emissive power even with a small amount of heating, as we want to avoid a large deviation from the theoretical estimation, where the optical parameters are assumed to be identical to the values at room temperature. Using this novel approach, which bridges thermal and optical phenomena to measure the radiative heat loss, we were able to measure $\varepsilon$ of a single isolated nanostructure, e.g., a nanoribbon, for the first time, without the effect of a substrate.

As a calibration of our fin technique to extract thermal emissivity, we also conducted a similar experiment with a rectangular beam cross-section which have larger thickness (~10 μm) than that of the ribbons as well as the skin-depth (~1 μm), such that the thermal radiation would follow the incoherent (broadband) bulk-like emission spectrum and the emissivity is expected to be close to the bulk value of $SiO_2$ (~0.9 at room temperature). We extracted the emissivity using the fin model following the same procedure described above. Firstly, we designed the thick $SiO_2$ length for 10 μm thick beams. To determine the suitable beam lengths to emphasize the radiative heat loss, we estimated thermal conductance by heat conduction and radiation with an assumed emissivity of 0.9. Supplementary Figure 5 shows that radiative conductance would be significant relative to heat conduction when the beam length is around 800 μm or longer. Thus, we fabricated the suspended $SiO_2$ beams with various lengths, namely, 100, 400, 600, and 800 μm, as shown in Supplementary Fig. 6. With these various lengths, we measured thermal conductivity of the beams, as shown in Supplementary Fig. 7. We analysed temperature ratios between the heating and sensing sides, $\gamma$, at each length. At the smallest length (100 μm), we obtained thermal conductivity of 1.41 W m⁻¹ K⁻¹ at room temperature. At this length, the radiation is negligible (see Supplementary Fig. 5) and the measured thermal conductivity agrees well with the expected bulk value of $SiO_2$. This validates our heat transfer measurements. By using the thermal conductivity value of the shortest beam or the bulk $SiO_2$ (~1.4 W m⁻¹ K⁻¹), $\Delta$ was calculated as shown in Supplementary Fig. 7b. Using the same fin model that fits all the sample lengths with a single fitting parameter in the emissivity, we found the best fit with an emissivity value of 0.77 (±0.07) for the 7.5 μm wide and 10 μm thick beams. This value agrees well with the theoretical expectation. For example, Golyk et al.[18] estimated an emissivity of around 0.7 for a cylindrical rod with 5 μm radius. The small discrepancy is likely due to the different cross-sectional geometries

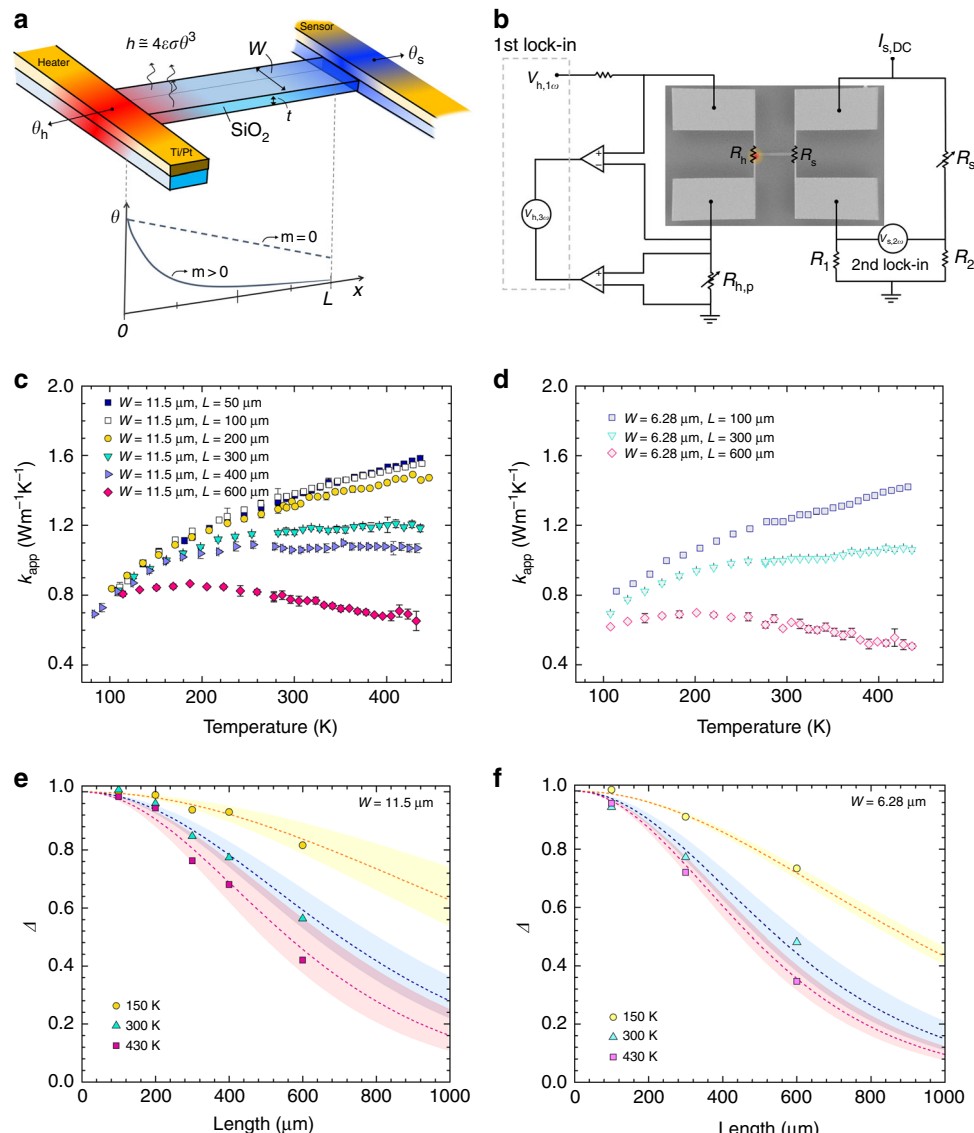

**Fig. 3** Thermal transport measurement methodology and results. **a** Schematic illustration of heat-transfer process in the suspended micro-device with $SiO_2$ nanoribbon overlaid with the colour code representing the temperature rise ($\theta$) on the heater, sensor, and nanoribbon. The schematic plot at the bottom presents profiles of $\theta$ along the ribbon: when there is no heat loss (the fin parameter $m = 0$), the $\theta$ profile is a straight line (dashed line), and when there is radiative heat loss ($m > 0$), $\theta$ is exponentially decayed along the length (solid line). **b** AC modulated heating scheme. The heating beam is heated by a $1\omega$ current, and the corresponding resistance change ($\Delta R_h$) due to the temperature rise is measured from the $3\omega$ voltage. The sensing beam resistance change, $\Delta R_s$, was measured from the $2\omega$ voltage using a small DC excitation current ($I_{s,DC}$). A Wheatstone bridge configuration was used on the sensing side to increase the measurement sensitivity. **c**, **d** Plots of measured apparent thermal conductivity ($k_{app}$) vs. temperature for nanoribbon widths of 11.5 μm (**c**) and 6.28 μm (**d**). $k_{app}$ was calculated using Eq. (6), which assumes a linear temperature profile along the ribbon (i.e., the dashed line in **a** with $m = 0$ or no radiation heat loss). The further below $k_{app}$ is from the true thermal conductivity of $SiO_2$, the larger the radiative heat loss. The error bars were determined by the standard deviations from the conductance measurement. **e**, **f** Plots of temperature ratios, $\Delta$, between the heating and sensing beams as a function of temperature. $\Delta$ is defined as $\gamma/\gamma_{lossless}$ and is a measure of the radiative heat loss. The experimental results (symbols) are compared to the fitted fin model (dotted lines), and the coloured area indicates the uncertainties of the fitted emissivity values

(rectangular in our experiment vs. cylindrical in Golyk et al.) as well as the surface roughness of our samples.

**Apparent thermal conductivity of individual $SiO_2$ nanoribbons.** Because amorphous $SiO_2$ ribbons were used, we could calibrate the new measurement technique on short samples (e.g., $L$ = 50 or 100 μm) with negligible radiative heat loss. $SiO_2$ is an ideal candidate to evaluate a new platform for heat-transfer measurement because it has very short phonon mean free paths (~1 nm at room temperature)[33,34]. Therefore, bulk-like thermal conductivity (1.3–1.4 W m$^{-1}$ K$^{-1}$ at 300 K) is maintained even at the nanometre

scale, as experimentally confirmed by us on $SiO_2$ nanotubes with shell thicknesses down to 7.7 nm[31]. For samples with longer length, $\gamma$ would be larger, and the measured apparent thermal conductivity would deviate from the intrinsic $\kappa$ of $SiO_2$, the difference in which is due to the radiative heat loss. Here, the apparent thermal conductivity is defined as $\kappa_{app} = G_{app}L/A_c$, where

$$G_{app} = G_b \frac{\theta_s}{\theta_h - \theta_s}. \tag{6}$$

When there is radiative heat loss ($m > 0$), $G_{app}$ in Eq. (6) deviates from the true $G_s$. Thus, by comparing the measured $\kappa_{app}$

to the intrinsic value for SiO$_2$, we can directly illustrate the importance of thermal radiation.

Figure 3c, d shows the temperature-dependent $\kappa_{app}$. Indeed, we observed saturated phonon thermal conductivity in a relatively short length up to 100 μm, which is the same as the bulk value. $\kappa_{app}$ clearly decreased with longer lengths as a result of the enlarged radiative heat loss, which becomes more pronounced at higher temperature. In general, $\kappa_{app}$ with various lengths converges to the bulk value at lower temperature, indicating the reducing radiative heat loss, as expected.

**Radiative behaviour of SiO$_2$ nanoribbons.** We compared the relative change among samples with various lengths covering the conduction- and radiation-dominant regimes by utilising the $m$ parameter, which quantifies the degree of radiative loss. Figure 3e, f presents the experimental and fitted $\Delta$ values based on the fin model with two different widths, 11.5 and 6.28 μm, respectively. A longer ribbon resulted in lower $\Delta$ because of the larger radiation heat loss relative to the heat conduction. However, $\Delta$ was close to ~1 for the short samples (e.g., $L = 50$ and 100 μm) because of the negligible radiative loss. Similarly, we plotted $\Delta$ at different temperatures, and, as expected, higher temperature generated lower $\Delta$. To extract $\varepsilon$, we directly compared the experimental and fitted $\Delta$ values. We used a single $\varepsilon$ value to fit all the samples with the same width but different lengths. The dashed line in Fig. 3 represents the $\Delta$ value based on the averaged $\varepsilon$ value showing the best fit over the entire length range,

and the shaded area indicates the uncertainty of the fitting. The uncertainty could originate from the slight variation in the actual widths of the fabricated samples with different lengths. Also, it may include the length-dependent behaviour with the nature of a broad range of propagating length as we shall see later. The temperature-dependent $\varepsilon$ values were determined by repeating the same fitting process for all the samples measured at different temperatures. To avoid any contribution from the surroundings, we employed triple radiation shields in a vacuum chamber to ensure the surrounding temperature is the same as the sample temperature.

Figure 4 presents the extracted $\varepsilon$ values at room temperature for ribbons with widths of 6.28 and 11.5 μm. In addition, the $\varepsilon$ of a 100-nm-thick film was analytically calculated to provide a baseline for comparison with the ribbons (Supplementary Figs. 8–10). Here, we observed two effects associated with the low dimensions in SiO$_2$ nanoribbons. First, the effective $\varepsilon$ of the ribbons was significantly lower than that of the bulk SiO$_2$ because the 100-nm thickness was much smaller than $\delta$ of SiO$_2$ (~1 μm at 300 K). However, more interestingly, a higher $\varepsilon$ was observed for the nanoribbons than for a film of the same thickness. Furthermore, a larger enhancement in $\varepsilon$ compared with the thin-film limit was observed for the ribbon with smaller width, 6.28 μm, which implies that a narrower width increases the resonance effect. These results can be explained by the localised resonant effect by SPhPs combined with the directional emission via the waveguided modes, which enhances $\sigma_{abs}$, as we shall discuss next.

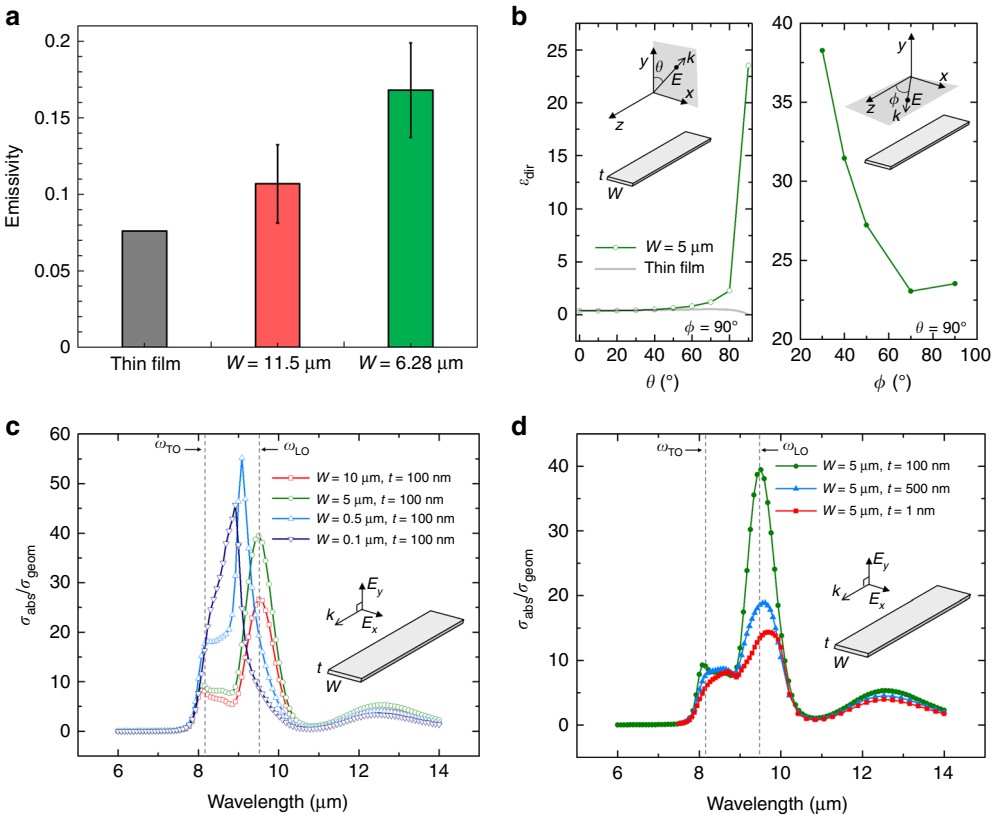

**Fig. 4** Enhanced emissivity with anisotropic nanoribbons. **a** Plots of the extracted emissivity at room temperature for ribbon widths of 6.28 and 11.5 μm, where the error bars are corresponding to the uncertainty in the fitting as shown in Fig. 3e, f. The grey bar represents the computed emissivity of an infinitely wide thin film of the same thickness as the ribbons, i.e., 100 nm. **b** Plots of directional emissivity ($\varepsilon_{dir}$) of a nanoribbon with $W = 5$ μm at a wavelength of 9.5 μm, where the incoming plane wave has propagation ($k$) directions where the polarisation directions are normal to $k$ in $x$–$y$ plane and $x$–$z$ plane, respectively for each $\varphi = 90°$ and $\theta = 90°$ cases, and its incident angle is controlled by $\theta$ and $\varphi$ as shown in the insets. **c, d** Enhanced absorption cross-sectional area as a function of wavelength, where the absorption cross-sectional area ($\sigma_{abs}$) is normalised by the geometrical cross-section ($\sigma_{geom}$) for various widths (**c**) and thicknesses (**d**)

**Comparison with the broadband background thermal emission**. We now analyse the contribution of the background broadband thermal emission, which is suppressed because of the small thickness (100 nm). In bulk $SiO_2$, the strong broadband absorption prevents the observation of further enhanced emission by polaritonic resonance. Therefore, a highly transparent (less absorbing) ribbon could signify the role of SPhPs in suppressing the broadband background, whose $\varepsilon$ was reduced to ~0.062 in the thin-film limit as shown in Fig. 4a. This significantly quenched background broadband $\varepsilon$ in the thin film provides the basis for dominantly coherent emission in the nanoribbon designs.

We now seek to understand the enhancement of the coherent emission in the nanoribbons, which was over 8.5-fold for the 6.28-μm-wide ribbon at 150 K compared with the thin-film limit calculated above. This enhancement is remarkable, considering that the spectral window for emission ($\omega_{TO}$ ~ 1251 cm$^{-1}$ to $\omega_{LO}$ ~ 1065 cm$^{-1}$ or from 8 to 9.5 μm) is now much narrower compared with the broad thermal spectrum. This enhancement can be understood from the strong resonant nature of the antenna-like nanoribbons, which results in an effective absorption cross-section ($\sigma_{abs}$) that is significantly higher than the geometrical one ($\sigma_{geom}$)[15]. Fernández-Hurtado et al.[15] suggested theoretical guidelines to overcome the far-field limit by Planck's law and have systematically studied the far field radiation heat transfer from and between various sub-wavelength dielectric objects through the theoretical and numeric modelling. The computational study introduced the enhanced far-field thermal conductance between two anisotropic objects by increasing absorption efficiency, which was subsequently verified experimentally by Thompson et al.[16]. Similarly, we modelled $\sigma_{abs}$ of the nanoribbons using a finite element method (FEM) approach in COMSOL.

Here, two plane waves propagating along the length with normal polarisation were simultaneously exposed to evaluate their averaged contribution, as shown in the inset in Fig. 4c. The modelling results of $\sigma_{abs}$, as observed in Fig. 4c, indeed showed strong enhancement within the Reststrahlen band (Fig. 4c). The results are clearly distinguished from the lossy dielectric regime at around 12 μm wavelength. The peak enhancement factors ranged from 25 to 55 in ribbons of different widths. The enhancement increased when the ribbon width was reduced from 10 μm to 500 nm. From our results on the dispersion relations of SPhPs in a 100-nm-thick $SiO_2$ (Fig. 1c), the wavelengths of the SPhPs ($2\pi/q$) ranged from 0.8 to 1.8 μm at around 9 μm. Therefore, the ribbons with widths down to 500 nm, which is larger than half of the shortest SPhP wavelength, can still support localised resonance SPhP modes.

Our modelling results also indicate that further reduction of the ribbon width below 500 nm would rather lead to smaller enhancement. For example, the peak enhancement factor for 100 nm width is 45, which is smaller than that of the 500 nm width (~55), see Fig. 4c. This difference occurs because when the width is reduced to less than half the wavelength of certain modes of SPhPs, the localised resonance mode can no longer be supported within the width of the ribbons. Therefore, as the width decreases, a greater portion of the localised SPhP modes become unavailable. This result highlights the importance of the anisotropic nanoribbon design with suitable widths in maximising the resonance effect and resultant coherent thermal emission, as an isotropic cross-section such as that in a cylindrical rod would be unable to support certain resonance modes and consequently lead to a much smaller emissivity (e.g., $\varepsilon = 0.025$ in a 100-nm-diameter nanorod[18]). The geometry-dependent peak shift in the metallic regime indicates the coherent effect, as opposed to the monotonic change in the dielectric regime (near 12 μm wavelength).

We also modelled the enhanced $\sigma_{abs}$ with different thickness. The thicker ribbon would result in the less overlaps of two evanescent waves at the top and bottom surfaces. Therefore, it yields smaller enhancement as shown in Fig. 4d. The enhanced $\sigma_{abs}$ is closely correlated with the enhancement of the measured $\varepsilon$. Therefore, we can attribute the largely enhanced $\varepsilon$ from the nanoribbons to the coherent emission within the Reststrahlen band caused by the strong SPhP resonance.

Similar to the analytical model using a thin film, we also conducted mode analysis study of nanoribbons with the finite width and thickness. We compared the propagating vector of the thin film to that of nanoribbons calculated using COMSOL. The ribbon structure assumes infinite length and the $k_p$ is parallel to the length. The two finite dimensions in thickness and width resulted in multiple modes, as shown in Fig. 5. The dominant modes with high $q$ are very close to the dispersion of the thin film. Therefore, similar to the analysis from the thin film, it can be concluded that symmetric configurations of electric fields at the surfaces dominantly support the energy confinement. In addition, we observed the electric field intensity distributions at the cross-section of nanoribbons. Indeed, it showed high intensity along the width. The lowest order of modes, marked with the red circle, are well overlapped with the fundamental mode available from the dispersion relation of a thin film. For higher order modes, we could clearly observe the coherent resonance along the width (Fig. 5b). Please note that still the propagating direction is along the length, and the diverse forms of polarisation increase the number of modes. Although the higher order modes have lower $q$ values, still multiple modes possess significantly higher $q$, compared to the light line. Further, the shift in $q$ is smaller with the narrower width (Fig. 5b). Corresponding to the real part ($q$) of propagating vector ($k_p = q + i\kappa$), we also studied the imaginary counterpart ($\kappa$). The propagating length was then calculated by 1/$2\kappa$, shown in Fig. 5c. A broad range of propagating length from 100s nm to ~1 mm, can be supported. It implies that the anisotropic structure could have the length-dependent emissivity. In our experiments (Fig. 3e, f), the fitted lines assume a constant $\varepsilon$ over all the lengths up to 600 μm. The error range covers the potential length-dependent behaviour. All modelling in this study has been done with an infinite length, but more rigorous theoretical study, such as fluctuational electrodynamics[15] could be a potential interesting study for the non-traditional far-field radiation.

In addition to the cross-sectional size-dependent emission, we modelled and compared the directional emissivity ($\varepsilon_{dir}$) of a nanoribbon and a film at 9.5-μm wavelength (e.g., $\omega_{LO}$ with the TE-polarised mode) using the finite-difference time-domain (FDTD) method, as shown in Fig. 4c. Compared with $\varepsilon_{dir}$ of the thin film (e.g., $\varepsilon_{dir}$ < 1 in all directions), $\varepsilon_{dir}$ of the nanoribbon with 5-μm width was distinguishably enhanced with the larger incident angle $\theta$ with fixed $\varphi = 90°$ (see the $\theta$ and $\varphi$ definition in the insets of Fig. 4c). The edges of the nanoribbon exposed to the incoming waves enhance the coupling of SPhPs, leading to the efficient emissive behaviour by the resonant modes. Similarly, as this nanoribbon is a 3D anisotropic structure, we also controlled $\varphi$ (Fig. 4c) and observed further enhancement when the incident wave was close to parallel to the length. The higher absorption originates from the infinite length of the nanoribbon used in the model. These findings clearly demonstrate the merit of the anisotropic designs to selectively emphasise the coherent modes. Our observation for the enhanced emissivity over the thin film limit originates from the fact that the selective enhancement (in the spatial and spectral domains) outweighs the reduced absorption by the smaller volume, compared to the skin-depth (determined by $\lambda/4\pi k$, where $k$ is the imaginary part of the complex refractive index). This point is different from the previous study by Thompson et al.[16]. As we observed from the angle-dependent absorption cross-section, the most significant

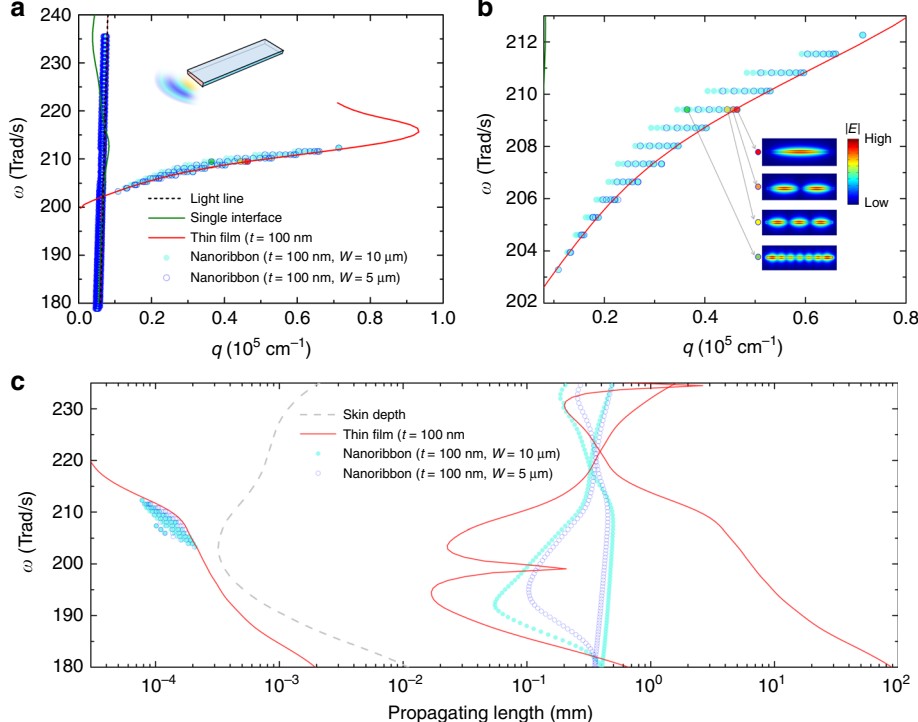

**Fig. 5** Numerical mode analysis of nanoribbons. **a** Dispersions of nanoribbons by numerical modelling. The numerical results (symbols) were compared to the analytical results of light line, single interface and a thin film. **b** Zoomed-in plots of (**a**). The insets represent the electric field intensity distributions in the cross-section of a 5-μm wide nanoribbon at the four colour-filled symbols. **c** Propagating lengths of nanoribbons with 5 and 10 μm widths, corresponding to the dispersion in (**a**, **b**)

enhancement occurred along the edge with sharper angles. It implies the strong directionality. Here, we want to clarify that the observed enhancement does not mean the super-Planckian emission. The spectral enhancement we observed from the modelling is up to 55, but it is within a certain spectral and directional range and thus is still insufficient to result in a super-Planckian hemispherical emissivity, which is an integrated effect over the entire spectral and angular ranges. This was also clearly shown by Fernández-Hurtado et al.[15]. Nevertheless, the fact that the emissivity is higher than the thin film limit underscores the prominent effect of the localised modes.

Our sensitive thermometry platform also enables the direct emissivity measurement at lower temperature, where the coherent effect is expected to be stronger but the emissive power is more challenging to measure (radiative power scales as $T^4$). At lower temperature, the phonon lifetime is expected to be longer and the $k_B T$ broadening is expected to be smaller, which will lead to a higher $Q$ factor of the emission. For the modelled emission from thin films, the spectral emittance showed a peak within the Reststrahlen band, whereas significantly suppressed emission was observed in the broad range. The modelling results assumed a temperature-independent optical parameter. Therefore, the temperature-dependent behaviour was only determined by the peak shift in the Planck's distribution and showed a maximal value near room temperature, corresponding to a thermal wavelength of ~10 μm (inset in Fig. 6a). For the nanoribbons, however, we observed higher $\varepsilon$ at lower temperature, up to 0.35 at 150 K for the 6.28-μm-wide ribbon (Fig. 6a). This unusual temperature dependence can be explained by the smaller $k_B T$, which increases the $Q$ factor because of the smaller loss (lower $\varepsilon''$) at lower temperature. The damping term in the dielectric function in the form of the Lorentzian function is correlated with the lifetime of the optical phonons, which is smaller at lower

temperature[27,35] and leads to less broadening of the emission peak and stronger coherent emission; thus, the emissive behaviour would be temperature dependent. To quantify the coherent emission, we introduce a coherent enhancement factor (CEF) determined by the ratio of $\varepsilon_{ribbon}$ to $\varepsilon_{film}$, where $\varepsilon_{ribbon}$ and $\varepsilon_{film}$ are $\varepsilon$ of a nanoribbon and film of identical thickness, respectively. As shown in Fig. 6b, the higher CEF observed at lower temperature is a very clear signature of the coherent mechanism[36–38]. The maximum CEF achieved here was 8.5 for the 6.28-μm-wide ribbon at 150 K.

In summary, we studied the coherent thermal emission in the far field via anisotropic dimension control of polaritonic nanoribbons. The combination of heat-transfer measurements and optical modelling provided a new platform to precisely engineer and quantify the coherent thermal emission of an individual nanoscale object. This developed technique enabled us to determine that the anisotropic polaritonic nanoribbon design provides an extra degree of freedom to realise a nanoscale coherent thermal emitter with no design limitation in terms of its length. The thin structure effectively suppressed incoherent emission in the broad range of the spectrum, and the controlled width of the nanoribbon enhanced the coherent emission mediated by SPhPs. Furthermore, we observed a higher CEF (up to 8.5) at lower temperature, which clearly emphasises the dominant coherent emission in the nanoribbon structures. The upper limit of the CEF can be further improved by engineering the dispersion of the emitter. Use of the grating effect by periodically arranging the nanoribbons[7,14] and/or metamaterials can lead to more confined surface modes by SPhPs with a higher $Q$ factor. In addition, the spatial distribution of coherent thermal emission can be controlled by selectively limiting polarisation. Such coherent thermal emitters with a selective spectral/spatial band could find wide-ranging applications for thermal

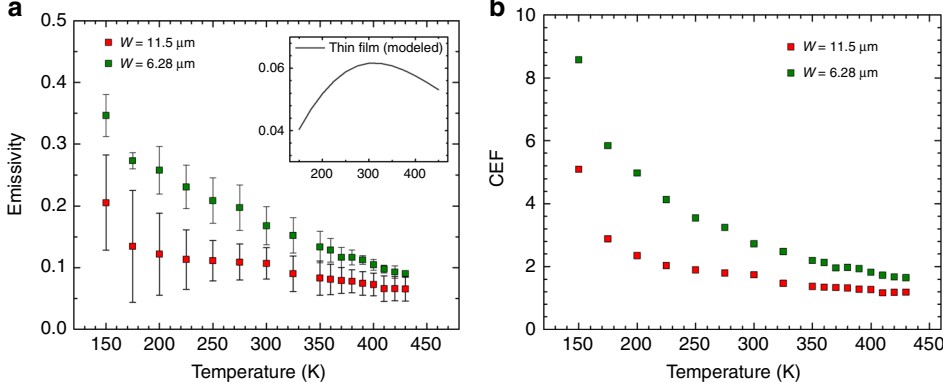

**Fig. 6** Temperature-dependent emissive behaviour. **a** Plots of emissivity as a function of temperature, where the error bars correspond to the uncertainty in the fitting as shown in Fig. 3e, f. The inset shows the modelled emissivity of an infinite thin film. **b** Plots of coherent enhancement factor (CER) as a function of temperature

management, energy conversion, bio/chemical sensing, light emission, and micro/nanomanufacturing[7,14,39–43].

## Methods

**Preparation and thermal measurement of SiO₂ nanoribbons**. To eliminate the potential impact of thermal contact resistance, the nanostructures were mono-lithically integrated with the thermal reservoirs. First, we prepared a 100-nm thermal oxide film on a Si wafer. Electrodes were patterned on top of the SiO₂ layer using photolithography, and a metal layer (Ti/Pt 80 nm) was deposited by sputtering. After defining the patterned area with Cr based on the designed nanoribbon structures, the SiO₂ layer was dry etched using CHF₃ and Ar gas (Oxford Plasmalab P80). Then, an additional Cr layer was deposited for protection with a larger area than that of the nanoribbons before the subsequent Si deep reactive ion etching (DRIE) by SF₆ gas (Oxford Plasmalab P100 RIE/ICP). This etching step provided a larger distance (~50 μm) between the Si substrate and SiO₂ nanoribbons. After the DRIE, the Cr layers were removed with a Cr etchant, and the Si was treated with a buffer oxide etchant (BOE), followed by XeF₂ etching to suspend the nanoribbons by removing the Si substrate underneath.

**Fin model**. A fin has high aspect ratios in dimensions to extend the surface area. Therefore, the temperature change along the length has the major contribution while it can be assumed that the temperature across the smaller dimension is constant. For this reason, the fin structure can possess the limited heat conduction and more effectively radiate heat, thus it has been widely used to cool devices as a heat sink[44].

The temperature profile can be described in Supplementary Fig. 2 by defining the temperature profile along the x-axis, parallel to the length, above the ambient temperature (θ) as below.

$$\theta = C_1 \cosh(mx) + C_2 \sinh(mx) \qquad (7)$$

where $C_1$ and $C_2$ are unknown constants.

$$\theta = T - T_0 \qquad (8)$$

The general solution introduces the feature of exponential temperature drop while propagating along the x-axis. The decaying rate is weighted by m, so-called fin parameter, and it is determined by

$$m = \sqrt{\frac{hP}{\kappa A_c}} \qquad (9)$$

$$h \cong 4\varepsilon\sigma T^3 \qquad (10)$$

where h is the heat transfer coefficient, ε is the emissivity, σ is the Stefan–Boltzmann constant, and T is the temperature. Quantifying the m parameter enables to obtain the ε.

To specify the unknown parameters from the solution for θ, here we summarise the boundary conditions for temperatures at x = 0 and x = L:

$$\theta|_{x=0} = C_1 = \theta_h \qquad (11)$$

$$\theta|_{x=L} = \theta_h \cosh(mL) + C_2 \sinh(mL) = \theta_s \qquad (12)$$

$$C_1 = \theta_h \qquad (13)$$

$$C_2 = \frac{\theta_s - \theta_h \cosh(mL)}{\sinh(mL)} \qquad (14)$$

where $\theta_h$ and $\theta_s$ are the temperature rise at the ends of the fin. By integrating two ends to the suspended metal beams, $\theta_h$ and $\theta_s$ are measured in this thermometry platform. Therefore, θ can be represented as

$$\theta = \theta_h \cosh(mx) + \frac{\theta_s - \theta_h \cosh(mL)}{\sinh(mL)} \sinh(mx). \qquad (15)$$

Next, we summarise the boundary conditions for heat flux at x = 0 and x = L. For heat flux at x = 0 (heating side),

$$Q_h = -\kappa A_c \frac{\partial\theta}{\partial x}\bigg|_{x=0} = -\kappa A_c m C_2. \qquad (16)$$

For heat flux at x = L (sensing side),

$$Q_s = -\kappa A_c \frac{\partial\theta}{\partial x}\bigg|_{x=L} = -\kappa A_c \{m C_1 \sinh(mL) + m C_2 \cosh(mL)\}. \qquad (17)$$

To meet the overall energy balance,

$$Q_s = G_b \theta_s \qquad (18)$$

$$-\kappa A_c \{m C_1 \sinh(mL) + m C_2 \cosh(mL)\} = G_b \theta_s \qquad (19)$$

$$-\kappa A_c m \left( \theta_h \sinh(mL) + \frac{\theta_s - \theta_h \cosh(mL)}{\sinh(mL)} \cosh(mL) \right) = G_b \theta_s \qquad (20)$$

$$\theta_h \sinh^2(mL) - \theta_h \cosh^2(mL) + \theta_s \left\{ \cosh(mL) + \frac{G_b}{\kappa A_c m} \sinh(mL) \right\} = 0 \qquad (21)$$

$$-\theta_h + \theta_s \left\{ \cosh(mL) + \frac{G_b}{\kappa A_c m} \sinh(mL) \right\} = 0 \qquad (22)$$

$$\theta_s = \frac{\theta_h}{\cosh(mL) + \frac{G_b}{\kappa A_c m} \sinh(mL)} \qquad (23)$$

$$\frac{\theta_s}{\theta_h}\bigg|_{m>0} = \frac{1}{\cosh(mL) + \frac{G_b}{\kappa A_c m} \sinh(mL)} \equiv \gamma. \qquad (24)$$

As shown in Supplementary Fig. 2c, the temperature profile become linear in the absence of heat loss (h = 0, and eventually m = 0).

$$Q_s = \frac{Q_h}{1 + \frac{G_b}{\kappa A_c} L} = \frac{Q_h}{1 + \frac{G_b}{G_s}} = \frac{G_s}{G_s + G_b} \theta_h \qquad (25)$$

$$G_s(\theta_h - \theta_s) = G_b \theta_s \qquad (26)$$

$$\theta_s = \frac{G_s \theta_h}{G_b + G_s} \qquad (27)$$

$$\frac{\theta_s}{\theta_h}\bigg|_{m=0} = \frac{G_s}{G_b + G_s} \cong \frac{G_s}{G_b} \equiv \gamma_{lossless}. \qquad (28)$$

Finally, we can define the ratio factor, Δ as

$$\Delta \equiv \frac{\gamma}{\gamma_{lossless}} = \frac{G_a}{G_s} \equiv \frac{\kappa_a}{\kappa_s} \qquad (29)$$

where $G_a$ is the measured apparent thermal conductance. By comparing the expected ratios of γ to the ratios of the measured conductivity, ε can be extracted.

**Heat transfer measurements**. Nanoscale emitters with radiative heat loss generate very small signals. Therefore, we employed an AC-modulated joule heating and lock-in thermometry technique, as depicted in Fig. 3b, to improve the sensitivity. In addition, we used a Wheatstone bridge to measure the temperature at the sensing side with a high resolution down to ~0.25 pW K$^{[-1}$ [32]. By measuring the RMS of the voltages at the heating and sensing sides corresponding to the modulated heating with $1\omega$ angular frequency (i.e., $V_{h,3\omega}$ and $V_{s,2\omega}$, respectively), the temperature rise could be determined[32]:

$$\theta_h = 3 \frac{V_{h,3\omega}}{I_\omega} \left(\frac{dR_h}{dT}\right)^{-1} \tag{30}$$

$$\theta_s = \sqrt{2} \frac{V_{s,2\omega}(R_s + R_{s,p} + R_1 + R_2)}{I_{s,DC} R_2} \left(\frac{dR_s}{dT}\right)^{-1}. \tag{31}$$

Here, $I_\omega$ and $I_{s,DC}$ are the AC-modulated input current to heating and DC current to the sensing beam, respectively; $R_h$ and $R_s$ are the resistance of the heating and sensing beam, respectively; and $R_{s,p}$, $R_1$, and $R_2$ are the resistances of the Wheatstone bridge in Fig. 3b. Equations (30) and (31) are applicable in the low-frequency regime, where the heat penetration depth is much longer than the length of the suspended bridge (Supplementary Note 4)[45]. The measured temperature rise at the heating and sensing sides was inputted into Eq. (4) to obtain the parameter $\gamma$. $\gamma_{lossless}$ was determined from the short sample, which has negligible radiative loss compared with conduction by phonons. Finally, the experimentally measured $\Delta$ ($=\gamma/\gamma_{lossless}$) could be compared using Eq. (32) to extract the unknown parameter, $m$ in the fin model, which allows both $h$ and $\varepsilon$ to be determined:

$$\Delta = \frac{\frac{1}{\cosh(mL) + \frac{G_h}{\kappa A_c m}\sinh(mL)}}{\frac{G_s}{G_h + G_s}} \tag{32}$$

As the measurements were done in vacuum (~$10^{-6}$ Torr) and thus the convection and conduction of air are not important, $h$ is only contributed by radiation heat loss, namely, only determined by the emissivity value. For each width, we used one single fitting parameter ($\varepsilon$) to fit all the ribbon lengths from 100 to 600 μm (Fig. 3e, f). We also showed that the measured thermal conductivity values of the shortest ribbons (50 μm long ribbon in Fig. 3c) agree well with the bulk value of amorphous SiO$_2$ (see Supplementary Fig. 3). At room temperature, the measured value is ~1.4 W m$^{-1}$ K$^{-1}$. We note that at 50 μm length, the radiation heat loss is negligible compared to heat conduction along the ribbon, as the calculated $\Delta$ is very close to 1 even if the emissivity is 0.90 (see Supplementary Fig. 4). The thermal conductivity of amorphous SiO$_2$ is well known and is not expected to be influenced by the size range studied here (down to ~100 nm thickness in this case) because of the short phonon MFP in SiO$_2$ (<10 nm), as well documented in the literatures[34,46]. We have also previously shown bulk-like thermal conductivity of SiO$_2$ nanostructures (nanowires and nanotubes) with thickness down to ~7 nm. Therefore, we obtained bulk-like thermal conductivity of the ribbons (directly from the shortest, 50 μm long ribbon, and indirectly from the fitting to all the other ribbons) over the entire temperature range of 150–430 K. We also obtained bulk thermal conductivity value from short (100 μm) and thick (10 μm) SiO$_2$ beams (see Supplementary Fig. 7). These results validate our heat conduction measurements. In particular, the use of monolithic devices eliminates any potential thermal contact resistance between the ribbons and the thermal reservoirs, which have been a challenging issue in many nanowire thermal transport measurements using similar platforms.

**Directional emissivity modelling**. The emissivity of an infinite ribbon was estimated from the absorption cross-section, which was calculated numerically using FDTD simulation (Supplementary Fig. 11). The ribbon was illuminated by a plane wave (emitted by a total-field scattered-field source[47], the profile of which in the frequency domain forms a Gaussian pulse, covering the frequency range in which SiO$_2$ has an appreciable extinction coefficient (55–280 Trad s$^{-1}$).

The incidence angle was manipulated to cover a hemisphere by changing the ribbon orientation angle, $\theta$, from 0° to 180° and the plane-wave travel direction, $\phi$, from 0° to 90° (see Supplementary Fig. 11). Because of the infinite length of the ribbon, we could accommodate the out-of-plane travel direction of the incident waves by applying Bloch periodic boundary conditions on the solution domain sides perpendicular to the direction along the length[47]. For incidence waves with $\phi \neq 90°$, we limited the frequency spectrum of the Gaussian plane wave, such that the deviation in the angle $\phi$ is below 5° for waves with different frequencies. The power absorbed by the ribbon was calculated by monitoring the net power flux crossing the boundary surrounding the ribbon in the time domain and then performing a Fourier transform to recover the absorbed power in the frequency domain. The spectral absorption cross-section was then calculated as the ratio between the absorbed spectral power and the plane-wave spectral source intensity. According to Kirchoff's law, the directional and spectral emissivity, $\varepsilon_{dir}$, has the same value as the spectral absorption efficiency, which can be calculated from the ratio between the absorption cross-section, $\sigma_{abs}$, and ribbon projected area, $\sigma_{geom}$ (assuming a ribbon of unit length)[48]:

$$\varepsilon_{dir}(\omega, \theta, \phi) = \frac{\sigma_{abs}(\omega, \theta, \phi)}{\sigma_{geom}(\theta, \phi)} \tag{33}$$

$$\sigma_{geom}(\theta, \phi) = (W \cos\theta + t \sin\theta) \sin\phi \cdot \tag{34}$$

## Data availability

The data that support the findings of this study is available from the corresponding authors upon reasonable request.

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

## Acknowledgements

The authors acknowledge Prof. Zhaowei Liu and Li Chen (UC San Diego) for their helpful comments and discussions. R.C. and S.S. acknowledge support from the National Science Foundation (DMR #1508420). Device fabrication was performed at the San Diego Nanotechnology Infrastructure (SDNI) of UCSD, a member of the National Nanotechnology Coordinated Infrastructure (NNCI), which is supported by the National Science Foundation (Grant ECCS-1542148). M.E. and R.P. acknowledge support by the Laboratory Directed Research and Development Program of Lawrence Berkeley National Laboratory under U.S. Department of Energy Contract No. DE-AC02-05CH11231. Work at the Molecular Foundry was supported by the Office of Science, Office of Basic Energy Sciences, of the U.S. Department of Energy under Contract No. DE-AC02-05CH11231. This research used resources of the National Energy Research Scientific Computing Center (NERSC), a U.S. Department of Energy Office of Science User Facility operated under Contract No. DE-AC02-05CH11231. This research used the Lawrencium computational cluster resource provided by the IT Division at the Lawrence Berkeley National Laboratory (Supported by the Director, Office of Science, Office of Basic Energy Sciences, of the U.S. Department of Energy under Contract No. DE-AC02-05CH11231).

## Author contributions

S.S. fabricated the devices, performed the heat-transfer measurements, and modelled the absorption cross-section and mode analysis using COMSOL Multiphysics. M.E. calculated the directional emissivity of a thin film and nanoribbon using FDTD. R.C. and R.P. supervised the project. All the authors contributed to the discussion of the results and wrote the manuscript.

## Additional information

**Competing interests:** The authors declare no competing interests.

