## [Peer Review File · Nature Communications]

Reviewers' Comments:

Reviewer #1:

Remarks to the Author:

The manuscript from Shin et al. presents an experimental study the far field thermal emission of nano ribbons and investigate the role of polarities resonance.

The findings are interesting and the novel and shed a light on a very subtle effect that has been not properly investigate up to now. Before recommending for publication in Nature Communications, I want to share with the authors a question that i believe it is important to address.

All the claims of the papers are based on indirect measurement of the radiative heat transfer: the authors measure thermal conduction through the nano ribbon and they claim that the difference with pure conduction is due to enhanced thermal radiation.

A claim based on such an evidence is then solid only when the control experiments are very well performed and presented.

It is my idea that this not exactly the case for this paper.

More in particular, authors assume that the Fin model for thermal conduction is valid but they do not present a clear experimental evidence of this claim. They should perform measurements with ribbons that are not expected to present thermal emission mediated by polaritons resonance and compare to the theoretical value.

The only data that are close to this are presented in figure 3 when they show the apparent thermal conductivity for samples with different length and they observe that long ribbons deviate from short one because of the increased role of thermal radiation. However no comparison to the fin model is presented for very short ribbon. If the comparison with the expected value of thermal conductivity were good than the claim could be considered more solid.

Similarly i believe that additional experiments are needed: measurements with thicker sample (thickness larger than the depth length) should be in agreement with Fin model because of the negligible role of polaritons mediated thermal radiation; further other kind of materials should be investigated (again material not presenting polaritonic resonance)

I am well aware that these measurements are complicated and surely time consuming. However considering the claim of the paper, I truly believe that the burden of proof is on the authors.

Once these concerns will be addressed I will be very glad to recommend publication in nature communications

Reviewer #2:

Remarks to the Author:

The authors present an experimental study of the thermal emissivity of silica nanoribbons. To be precise, by using a novel experimental platform, the authors are able to extract the information of the thermal emissivity of SiO₂ slabs with a thickness of 100 nm, which is much smaller than the thermal wavelength given by Wien's displacement law. The main conclusion of this work is the fact that these nanoribbons exhibit a larger emissivity than a thin film of the same thickness, although in all cases the emissivity is smaller than 1 (i.e., the total thermal emission is smaller than that of a black body of the same dimensions). The authors attribute this enhanced emissivity (as compared to a thin film) to the coherent resonant effect of surface phonon polaritons (SPhPs) in these systems that results in a very anisotropic thermal emission.

To my knowledge, this work presents one of the few measurements of the far-field thermal emission of a single subwavelength object. These measurements, in turn, show the dramatic

failure of Planck's law to correctly describe the thermal emission properties of such small objects. Moreover, the experimental platform is novel and very ingenious. So, for all these reasons, there is little doubt that the experimental results presented in this manuscript are of great interest for the field of thermal radiation and they deserve publication in a high profile journal like Nature Communications. However, the interpretation of the data and the corresponding theoretical analysis are not at the same level, and some aspects are clearly incorrect (see below). Thus, I think that the manuscript must be thoroughly revised to provide a correct interpretation of the experimental results, which requires a much more rigorous theoretical analysis than what is currently presented in this manuscript.

Let me first explain what the major problem with the interpretations is. The main finding of this work is that the silica nanoribbons exhibit an enhanced emissivity as compared to a thin film of the same thickness. The proposed explanation for this observation is that these nanoribbons emit very efficiently along the edges, which is somehow related to the coherent effect that results from the contribution of SPhPs. This last part of the proposed mechanism is incorrect. The far-field thermal emission (and radiative heat transfer) of silica nanoribbons and silica suspended pads, very similar to the ones studied in this work, has been theoretically studied in Ref. 34. There it has been shown that these nanoribbons exhibit a very anisotropic thermal emission and, in particular, the directional emissivity along the edges is extremely efficient and, in particular, it largely overcomes the blackbody limit. It is this peculiar directional emissivity what results in a total emissivity being larger than in the case of thin films. Moreover, Ref. 34 shows that the origin of the huge directional emissivity along the edges can be traced back to the nature of the waveguide modes that are sustained by these dielectric structures and, in particular, it is shown that the TE waveguide modes are the ones that dominate the absorption/emission properties of these systems. So, therefore, the SPhPs, which are surface TM modes, are not the most relevant modes in this problem. Moreover, even the TM modes sustained by these nanoribbons cannot be considered, in general, as SPhPs (this only could make sense in the frequency region where the real part of the dielectric constant becomes negative, which is not the only one that contributes to the thermal emission). It makes much more sense to talk about the guiding modes sustained by these structures, rather than about surface modes. So, in short, the explanation of the enhanced emissivity in terms of SPhPs is simply incorrect. This is a major issue since, in particular, the authors argue that they designed their structures based on the analysis of the SPhPs.

There are many more issues that the authors should amend and address before the paper can become publishable. In what follows, I list them following the order in which they appear in the manuscript:

1) Already in the abstract the authors talk about coherent thermal emission. This term is rather confusing because the coherence of the radiation is not analyzed or probed in these experiments. Actually, what the authors seem to mean is narrowband thermal emission (i.e. more monochromatic), as opposed to the broadband emission of a black body. I would personally change this because it is very misleading (it took me a while to understand what the authors meant by coherent).

2) When Fig. 1 is first discussed in the introduction, it is unclear how the emissivity is computed. This is a non-trivial issue since notice that the authors do not provide results for the total emissivity, which is actually the most important quantity in this work. This only becomes clear much later in the manuscript. Moreover, the meaning of the angle theta has to be somehow guessed since it is not indicated in that figure. Again, this only becomes clear later in the manuscript.

3) In page 6, the authors discuss the results by first analyzing the SPhPs, see Eq. 1. As explained above, these are not the most relevant modes for the thermal emission of these nanoribbons. Neglecting the TE modes, the authors are ignoring a very important part of the underlying physics.

Moreover, Eq. 1 is not well explained in the text and it is not obvious whether it applies to the system under study. First, the authors should clearly explain that this is the dispersion relation of TM modes for an infinite slab. But, is it obvious that the infinite slab approximation is justified in this case? The width of the nanoribbons is not that large. Second, this expression is taken from a work where the SPhPs of a 2D material were studied. Is it obvious that this formula applies to a system with a thickness of 100 nm? The properties of those modes (and TE modes as well) can be described with standard theories of dielectric waveguides (see e.g. Ref. 34 for the case of an infinite slab).

4) The term “thermal fin model” is not so well-known. The authors should provide a reference or refer to the Methods section.

5) In page 10 the authors state that: “As the lengths of the ribbons were much longer than the penetration depth³⁴, we assumed a single ϵ value to fit all the samples with the same width but different lengths”. This is not quite correct. In Ref. 34 it is explained that depending on the frequency (in the range relevant for the thermal emission), the propagation length can be larger than 1 mm. Thus, this assumption has to be justified with an analysis of the corresponding modes or, ideally, with a theoretical analysis of the length-dependence of the emissivity.

6) The most important result is the observation of the enhanced emissivities. This observation, in turn, shows the failure of the classical Planck’s law. But then, the obvious question is whether these results can be explained by fluctuational electrodynamics, which is the rigorous theory describing any thermal radiation problem, irrespective of the size of the objects involved. This is not addressed at all in this work. The calculation of the total emissivity of silica nanoribbons has been provided in Ref. 34. Would it be possible to present similar results for the exact dimensions of these experiments? I understand that the nanoribbons are very long and their modelling may be challenging, but it is worth trying since this is, in my opinion, the most important question related to the main results of this work. If it is not possible, the authors should at least mention that this is an interesting open problem.

7) The main argument used to explain the main observation is basically borrowed from Ref. 34. In this sense, I think that it is fair to refer to this work in a more explicitly way. Otherwise, one may give the impression that this explanation is proposed for the first time in this work.

8) When the authors discuss the direction dependence of the absorption cross section, they actually show results for the TE polarization (see Fig. 4c). These results clearly demonstrate that the TE modes play a crucial role and, in turn, contradict the central conclusion that the SPhPs dominate the physics. In other words, those results should have led the authors to conclude that their explanation is not consistent.

Reviewer #3:

Remarks to the Author:

In this work, Shin et al. demonstrated an experimental platform to quantify the thermal radiation from individual nanoribbons. Using this platform, they quantified the size dependence of the emissivity of nanoribbons. The authors designed the experiment well, and the contribution of the radiative heat transfer is extracted in a well-controlled way. The emissivity is obtained from the contribution of radiation. The authors were intended to show that ribbons that can support thermally excited localized phonon polaritonic resonances can create coherent thermal emission and result in an emissivity that is larger than the thin-film counterpart or ribbons that do not support localized resonances. Although I like the technique and the platform of this work, I think the analysis of the phonon polaritons in the ribbon and the connection with the measured emissivity are not clear, and therefore, the intended demonstrations were not convincing. I cannot

recommend its acceptance in its current form.

1. Equation (1) is quoted from Ref. 27, which is for a thin hBN film. hBN is a hyperbolic material, and its waveguide dispersion may not be directly applicable for SiO₂ ribbons. For SiO₂ in its Reststrahlen band, the authors may want to treat it as a thin metallic film and consider the coupled surface phonon polariton, which has a different dispersion from Eq. (1) in this work. The wavelength results shown in Fig. 1c, therefore, cannot be trusted. Also, what is the value of l ? Based on Eq. (1), shouldn't there be more than one order of modes?
2. The authors may want to provide information on the propagation length of the surface phonon polaritons. This is important to understand the size dependence of the resonance. If the propagation length is too smaller, then the ribbon width would not matter.
3. The author may need to provide a model to describe the geometric dependence of the ribbon resonances. After all, this work is aimed to demonstrate coherent thermal emission. Without such a model, it is hard to connect the total radiation loss in a broad wavelength and angular range with coherent thermal emission.
4. The author may want to double check the unit used for intensity in Fig. 1d.
5. The ribbon is not a Super-Planckian emitter, as Ref. 34 has proved. Therefore, it is not entirely obvious that why a ribbon should have a high hemispherical emissivity, which is an integrated effect, if it can support a localized resonance only at a given wavelength and a given direction. The authors may need to prove it at least numerically.

We thank all reviewers for their positive evaluation on our results and valuable comments. Based on these comments, we have improved the interpretation of our results with supported theoretical models of more direct mode analysis for the nanoribbon structures, and we have carried out additional experiments. In broad categories, we supplemented our manuscript with: 1) fin model verification using thick SiO₂ rods, 2) dominant mode analysis and 3) geometric dependency analysis. The detailed responses are below for each of the comments.

Color codes used in this response letter:

Blue Italic: original review comments;

Black: our responses;

Red: revisions made in the manuscript.

Reviewer #1:

The manuscript from Shin et al. presents an experimental study the far field thermal emission of nano ribbons and investigate the role of polarities resonance.

The findings are interesting and the novel and shed a light on a very subtle effect that has been not properly investigate up to now. Before recommending for publication in Nature Communications, I want to share with the authors a question that i believe it is important to address.

All the claims of the papers are based on indirect measurement of the radiative heat transfer: the authors measure thermal conduction through the nano ribbon and they claim that the difference with pure conduction is due to enhanced thermal radiation.

A claim based on such an evidence is then solid only when the control experiments are very well performed and presented.

It is my idea that this not exactly the case for this paper.

More in particular, authors assume that the Fin model for thermal conduction is valid but they do not present a clear experimental evidence of this claim. They should perform measurements with ribbons that are not expected to present thermal emission mediated by polaritons resonance and compare to the theoretical value.

The only data that are close to this are presented in figure 3 when they show the apparent thermal conductivity for samples with different length and they observe that long ribbons deviate from short one because of the increased role of thermal radiation. However no comparison to the fin model is presented for very short ribbon. If the comparison with the expected value of thermal conductivity were good than the claim could be considered more solid.

Similarly i believe that additional experiments are needed: measurements with thicker sample (thickness larger than the depth length) should be in agreement with Fin model because of the negligible role of polaritons mediated thermal radiation; further other kind of materials should be investigated (again material not presenting polaritonic resonance)

I am well aware that these measurements are complicated and surely time consuming. However considering the claim of the paper, I truly believe that the burden of proof is on the authors.

Once these concerns will be addressed I will be very glad to recommend publication in nature communications

Response:

We thank the reviewer for his/her positive comments on our experimental study, and for the valuable suggestion of additional control experiments for the emissivity value extraction based on the fin model. Our responses to this question can be summarized in two parts: (i) We would like to clarify that the fin model was applied to all the samples of various length for each of the width values, as shown in Fig. 3(e) and (f). (ii) Following the suggestion by the reviewer, we performed additional experiments with large SiO₂ ribbons (7.5 μm wide and 10 μm thick) with various lengths, and obtained bulk-like emissivity by following the same fin model, as expected. This further validates our methodology of extracting emissivity from the fin model. The detailed responses are as follow.

First, at each temperature, the fin model was applied and fit to the measured “Δ” of ALL the nanoribbon samples of various lengths ranging from 100 to 600 μm for each width (either the 11.5 μm width, see Fig. 3e, or 6.28 μm, see Fig. 3f). Here, based on Eqs 4 and 5 in the manuscript,

$$\Delta = \gamma / \gamma_{lossless} = \frac{\frac{1}{\cosh(mL) + \frac{G_b}{\kappa A_c m} \sinh(mL)}}{\frac{G_s}{G_b + G_s}},$$

where $\Delta = \sqrt{\frac{hP}{kA_c}}$. Therefore, the measured “ Δ ” only depends on heat transfer coefficient (h) between the ribbon and the environment if the intrinsic thermal conductivity (k) of the ribbon is known. As the measurements were done in vacuum ($\sim 10^{-6}$ Torr) and thus the convection and conduction of air is not important, h is only contributed by radiation heat loss, namely, $h \cong 4\varepsilon\sigma T^3$. For each width, we used one single fitting parameter (ε) to fit all the ribbon lengths from 100 to 600 μm (Fig. 3e and 3f). The equation with one single parameter can fit well with all the ribbon lengths, suggesting that the use of the intrinsic thermal conductivity value is validated.

As the reviewer suggested, we compare the measured thermal conductivity of the shortest ribbons to the bulk value. The thermal conductivity of the 50 μm long ribbon (data in fig. 3c, e.g., ~ 1.4 W/m-K at room T) agrees well with the literature value for amorphous SiO_2 within 10% error range, The comparison is shown in Fig. S3 below. Therefore, we in fact showed that *“comparison to the fin model is presented for very short ribbon”*, and also showed that *“the comparison with the expected value of thermal conductivity were good”*. We can also theoretically show that at 50 μm length, the radiation heat loss is negligible compared to heat conduction along the ribbons, as the calculated Δ would be very close to 1 even if the emissivity were 0.90. Also see Fig. S4 below for calculated Δ vs length for ribbon width of 10 μm and thickness of 100 nm.

Figure S3 Measured thermal conductivity of the shortest nanoribbon ($W=11.5 \mu\text{m}$, $L=50 \mu\text{m}$ and $t=100$ nm) were compared to the bulk SiO_2 thermal conductivity.

Figure S4. Plots of the modeled temperature rising ratio, Δ with various lengths at room temperature

Besides being a material that supports SPhP, thermal SiO₂ was chosen in this study also because its thermal conductivity is well known and is not expected to be influenced by size (~100 nm thickness in this case) because of the short phonon MFP in SiO₂ (<10 nm), as well documented in the literatures [References: Yang et al., Ballistic phonon penetration depth in amorphous silicon dioxide, *Nano Lett.* 17, 7218 (2017); Goodson et al., Prediction and measurement of the thermal conductivity of amorphous dielectric layers. *Journal of Heat Transfer*, 116, 317 (1994)]. We have also previously shown bulk-like thermal conductivity of SiO₂ nanostructures (nanowires and nanotubes) with thickness down to ~7 nm (Kwon et al., Unusually High and Anisotropic Thermal Conductivity in Amorphous Silicon Nanostructures, *ACS Nano*, 2017, 11 (3), pp 2470–2476). Therefore, we obtained bulk-like thermal conductivity of the ribbons (directly from the shortest, 50 μm long ribbon, and indirectly from the fitting to all the other ribbons) over the entire temperature range of 150 to 430 K. As shown later, we also obtained bulk thermal conductivity value from short (100 μm) and thick (10 μm) SiO₂ beams that we recently fabricated and measured during this revision process (see Fig. S7). These results validate our heat conduction measurements. In particular, the use of the monolithic devices eliminates any potential thermal contact resistance between the ribbons and the thermal reservoirs, which have been a challenging issue in many nanowire thermal transport measurements using similar platforms.

We made the following changes to the manuscript to better clarify the application of the fin model and the validation of our heat transfer measurements.

In the Supplementary Information, add Figure S4:

Figure S4. Plots of the modeled temperature rising ratio, Δ with various lengths at room temperature

On page 23, add:

“As the measurements were done in vacuum ($\sim 10^{-6}$ Torr) and thus the convection and conduction of air is not important, h is only contributed by radiation heat loss, namely, only determined by the emissivity value. For each width, we used one single fitting parameter (ϵ) to fit all the ribbon lengths from 100 to 600 μm (Fig. 3e and 3f). We also showed that the measured thermal conductivity values of the shortest ribbons (50 μm long ribbon in Fig. 3c) agree well with the bulk value of amorphous SiO_2 (see Fig. S3). At room temperature, the measured value is ~ 1.4 W/m-K. We note that at 50 μm length, the radiation heat loss is negligible compared to heat conduction along the ribbon, as the calculated Δ is very close to 1 even if the emissivity is 0.90 (see Fig. S4). The thermal conductivity of amorphous SiO_2 is well known and is not expected to be influenced by the size range studied here (down to ~ 100 nm thickness in this case) because of the short phonon MFP in SiO_2 (< 10 nm), as well documented in the literatures^{46,47}. We have also previously shown bulk-like thermal conductivity of SiO_2 nanostructures (nanowires and nanotubes) with thickness down to ~ 7 nm. Therefore, we obtained bulk-like thermal conductivity of the ribbons (directly from the shortest, 50 μm long ribbon, and indirectly from the fitting to all the other ribbons) over

the entire temperature range of 150 to 430 K. We also obtained bulk thermal conductivity value from short (100 μm) and thick (10 μm) SiO₂ beams (see Fig. S7). These results validate our heat conduction measurements. In particular, the use of the monolithic devices eliminates any potential thermal contact resistance between the ribbons and the thermal reservoirs, which have been a challenging issue in many nanowire thermal transport measurements using similar platforms.”

We revised the figure caption of fig. 3(e) and (f) to clearly mention that the dotted lines are from the fin model.

Figure 3. e–f, Plots of temperature ratios, Δ , between the heating and sensing beams as a function of temperature. Δ is defined as $\gamma/\gamma_{lossless}$ and is a measure of the radiative heat loss. The experimental results (symbols) are compared to the fitted fin model (dotted lines), and the coloured area indicates the uncertainties of the fitted emissivity values.

Now we turn to the second point: validating of the emissivity measurement using the fin model. As the reviewer suggested, we also conducted additional experiments with thick SiO₂ samples where the thickness is larger than the skin-depth, so the thermal radiation would follow the incoherent (broadband) bulk-like emission spectrum. As a result, we would expect the extracted emissivity value to be close to the bulk value of SiO₂ (~0.9 at room temperature). In our new experiments, we extracted the emissivity using the fin model, by following the same procedure as we did for the thin nanoribbons.

We designed the thick SiO₂ beams with 10 μm thickness, i.e., comparable with the skin depth of infrared from 8-10 μm in bulk SiO₂. To determine the suitable beam length to emphasize

the radiative heat loss, we estimated the thermal conductance by conduction and radiation with an assumed emissivity of 0.9.

Figure S5. Plots of calculated thermal conduction due to phonon conduction and radiation heat loss as a function of the beam length for beam width of $7.5 \mu\text{m}$ and thickness of $10 \mu\text{m}$.

Figure S5 shows that radiative conductance would be significant relative to heat conduction when the beam length is around $800 \mu\text{m}$ or longer. Thus, we fabricated the suspended SiO_2 beams with various lengths, ranging from 100 to $800 \mu\text{m}$ as shown in Fig. S6.

Figure S6. A SEM image of a suspended long SiO₂ beam with 10 μm thickness and 7.5 μm width.

With various lengths, namely, 100, 400, 600 and 800 μm, we measured thermal conductivity as shown in Fig. S7. We analyzed temperature ratios between heating and sensing sides, Δ , at each length. At smallest length (100 μm), we obtained thermal conductivity of 1.41 W/m-K at room temperature. Again, at this length, the radiation is negligible (see Fig. S5) and the measured thermal conductivity agrees well with the expected bulk value of SiO₂. This again validates our heat transfer measurements. By using the thermal conductivity value of the shortest beam or the bulk SiO₂ (~1.4 W/m-K), Δ was calculated as shown in Fig. S7. Using the same fin model that fits all the sample lengths with a single fitting parameter (i.e., the emissivity), we found the best fit with an emissivity value of 0.77 (± 0.07) for the 7.5 μm wide and 10 μm thick beams. This value agrees well with the theoretical expectation. For example, Golyk *et al.* estimated around an emissivity of 0.7 for a cylindrical object with 5 μm radius [Ref: V. A. Golyk *et al.*, Heat radiation from long cylindrical objects, Phys. Rev. E 85, 046603 (2012)].

Figure S7. Plots of (a) apparent thermal conductivity at room temperature and (b) Δ with various lengths of samples, where the best fit emissivity is 0.77 and the fitting has a standard deviation of 0.07 in absolute emissivity.

On Page 9, we added:

“As a calibration of our fin technique to extract thermal emissivity, we also conducted a similar experiment with a rectangular beam cross section which have larger thickness ($\sim 10\ \mu\text{m}$) than that of the ribbons as well as the skin-depth ($\sim 1\ \mu\text{m}$), such that the thermal radiation would follow the incoherent (broadband) bulk-like emission spectrum and the emissivity is expected to be close to the bulk value of SiO_2 (~ 0.9 at room temperature). We extracted the emissivity using the fin model following the same procedure described above. Firstly, we designed the thick SiO_2 length for $10\ \mu\text{m}$ thick beams. To determine the suitable beam lengths to emphasize the radiative heat loss, we estimated thermal conductance by heat conduction and radiation with an assumed emissivity of 0.9 . Figure S5 shows that radiative conductance would be significant relative to heat conduction when the beam length is around $800\ \mu\text{m}$ or longer. Thus, we fabricated the suspended SiO_2 beams with various lengths, namely, 100 , 400 , 600 , and $800\ \mu\text{m}$, as shown in Fig. S6. With these various lengths, we measured thermal conductivity of the beams, as shown in Fig. S7. We analyzed temperature ratios between the heating and sensing sides, γ , at each length. At the smallest length ($100\ \mu\text{m}$), we obtained thermal conductivity of $1.41\ \text{W/m-K}$ at room temperature. At this length, the radiation is negligible (see Fig. S5) and the measured thermal conductivity agrees well with the expected bulk value of SiO_2 . This validates our heat transfer measurements. By using the thermal conductivity value of the shortest beam or the bulk SiO_2 ($\sim 1.4\ \text{W/m-K}$), Δ was calculated as shown in Fig. S7b. Using the same fin model that fits all the sample lengths with a single fitting parameter in the emissivity, we found the best fit with an emissivity value of $0.77 (\pm 0.07)$ for the $7.5\ \mu\text{m}$ wide and $10\ \mu\text{m}$ thick beams. This value agrees well with the theoretical expectation. For example, Golyk et al.¹⁸ estimated an emissivity of around 0.7 for a cylindrical rod with $5\ \mu\text{m}$ radius. The small discrepancy is likely due to the different cross sectional geometries (rectangular in our experiment vs. cylindrical in Golyk et al.) as well as the surface roughness of our samples.”

We also appreciate the reviewer’s suggestion of testing a different material. We have attempted similar fabrication of polymer beams with large cross section, but it turned out unsuccessful due to the difficulty of finding a suitable fabrication recipe for an entirely different material within the short timeline (e.g., for selective etching to release the beams). Nevertheless, we hope the reviewer would agree with us that our new measurements results on thick SiO_2 beams with bulk-like emissivity and thermal conductivity would provide sufficiently convincing calibration for our

measurement platform. We again appreciate this reviewer's strong recommendation of our work for publication.

Reviewer #2:

The authors present an experimental study of the thermal emissivity of silica nanoribbons. To be precise, by using a novel experimental platform, the authors are able to extract the information of the thermal emissivity of SiO₂ slabs with a thickness of 100 nm, which is much smaller than the thermal wavelength given by Wien's displacement law. The main conclusion of this work is the fact that these nanoribbons exhibit a larger emissivity than a thin film of the same thickness, although in all cases the emissivity is smaller than 1 (i.e., the total thermal emission is smaller than that of a black body of the same dimensions). The authors attribute this enhanced emissivity (as compared to a thin film) to the coherent resonant effect of surface phonon polaritons (SPhPs) in these systems that results in a very anisotropic thermal emission.

To my knowledge, this work presents one of the few measurements of the far-field thermal emission of a single subwavelength object. These measurements, in turn, show the dramatic failure of Planck's law to correctly describe the thermal emission properties of such small objects. Moreover, the experimental platform is novel and very ingenious. So, for all these reasons, there is little doubt that the experimental results presented in this manuscript are of great interest for the field of thermal radiation and they deserve publication in a high profile journal like Nature Communications. However, the interpretation of the data and the corresponding theoretical analysis are not at the same level, and some aspects are clearly incorrect (see below). Thus, I think that the manuscript must be thoroughly revised to provide a correct interpretation of the experimental results, which requires a much more rigorous theoretical analysis than what is currently presented in this manuscript.

Let me first explain what the major problem with the interpretations is. The main finding of this work is that the silica nanoribbons exhibit an enhanced emissivity as compared to a thin film of the same thickness. The proposed explanation for this observation is that these nanoribbons emit very efficiently along the edges, which is somehow related to the coherent effect that results from the contribution of SPhPs. This last part of the proposed mechanism is incorrect. The far-field thermal emission (and radiative heat transfer) of silica nanoribbons and silica suspended pads, very similar to the ones studied in this work, has been theoretically studied in Ref. 34. There it has

been shown that these nanoribbons exhibit a very anisotropic thermal emission and, in particular, the directional emissivity along the edges is extremely efficient and, in particular, it largely overcomes the blackbody limit. It is this peculiar directional emissivity what results in a total emissivity being larger than in the case of thin films. Moreover, Ref. 34 shows that the origin of the huge directional emissivity along the edges can be traced back to the nature of the waveguide modes that are sustained by these dielectric structures and, in particular, it is shown that the TE waveguide modes are the ones that dominate the absorption/emission properties of these systems. So, therefore, the SPhPs, which are surface TM modes, are not the most relevant modes in this problem. Moreover, even the TM modes sustained by these nanoribbons cannot be considered, in general, as SPhPs (this only could make sense in the frequency region where the real part of the dielectric constant becomes negative, which is not the only one that contributes to the thermal emission). It makes much more sense to talk about the guiding modes sustained by these structures, rather than about surface modes. So, in short, the explanation of the enhanced emissivity in terms of SPhPs is simply incorrect. This is a major issue since, in particular, the authors argue that they designed their structures based on the analysis of the SPhPs.

Response:

We appreciate the reviewer's positive evaluation on our experimental platform as a novel approach to measure far-field thermal radiation. Also, we thank the reviewer for his/her in-depth comments on the mechanisms related to radiation from the nanoribbon geometry. We significantly supplemented our study by adding more detailed mode analysis using analytical and numerical models, and we further studied potentially geometrical effects for the enhanced effective absorption cross-section considering both the TE and TM polarizations.

First of all, we would like to accept that the current paper is more focused on experiments to measure emissivity of suspended nanoribbons than a detailed modeling of the observed phenomena. The experiments conducted in this study are very challenging. Therefore, in that sense this is the first ever reporting of thermal emission from single nanoribbons and this methodology may serve as way to experimental measure thermal emission from other anisotropic nanosized objects such as suspended nanowires and nanotubes. The detailed modeling of the observed phenomenon including effect of varying length, width, thickness, temperature is by itself a mammoth task and will be undertaken by us in future studies, however in this paper with limited

modeling (which itself were time consuming which we feel the reviewers can relate to) we have tried to provide a mechanistic understanding of our experimental results.

Discussion on the mechanism of enhancement: The main finding in this paper is that a nanoribbon has higher emissivity than a thin film. Fernández-Hurtado et al. (Ref. 34 in the old version, and now Ref. 15) provided a detailed understanding of thermal radiation from thin films and other anisotropic geometries. In their study, they studied both: Emission along a given direction from thin films and also total emission along with a dispersion analysis in thin films (Figs. 7 and 8 in their paper). In this paper, we have developed experimental approach to observe far field hemispherical emission from a single nanoribbon. The difference between the paper by Fernández-Hurtado *et al.* and our current paper is to understand the emission enhancement in nanoribbon over thin film limits. Based on the study of thin films, Fernández-Hurtado *et al.* showed that their results can be explained based on guided waves in thin films, including both the leaky (left side of the light line) and evanescent guided modes (right side of the light line). In an infinite thin film, the dominant contribution to emissivity comes from both the leaky and guided modes, as clearly shown by Fernández-Hurtado et al. (Fig. 5 and Fig. 8 in their paper). In their paper (Fig. 5), the TE polarization has a much higher contribution to the directional emissivity compared to the TM polarization. In our own modeling result of 100 nm-thick thin films with infinite width (see figure below, Fig. S14), we also showed that, similar to the finding of Fernández-Hurtado *et al.*, the TE polarization has a larger contribution to the modeled hemispherical emissivity, and the spectral range for high hemispherical emissivity covers both the leaky and evanescent guided modes. Therefore, we fully agree with the reviewer that that TE guided modes play an important role in the emissivity of the thin film, and we have now included this important point in the introduction and other parts of our paper and properly cited the important and inspirational work by Fernández-Hurtado *et al.* (see details in our responses to *reviewer 2 – Q7*).

Figure S14. Plots of hemispherical emissivity of TE and TM mode of a 100 nm thin film.

In addition to TE, we hope the reviewer would also agree with us that the TM polarization also plays an important role in the modeled hemispherical emissivity of 100-nm thick SiO₂ thin films, even though its contribution is slightly lower than that of the TE polarization (see Fig. S14). We believe that this small difference between our results and that of Fernández-Hurtado et al. (in terms of the role of TE vs TM) could originate from the fact that we are looking at the hemispherical emissivity (all the angles). Furthermore, we calculated the dispersion of the thin films for all the three modes available in a thin film (please see our response to **Reviewer 2-Q3** and Figures S19 and S21 therein, and also the supplementary note S8 for the details of the dispersion calculations): **TE, TM-asymmetric, and TM-symmetric** (pls see details later). The first two were the same as the ones calculated in Fernández-Hurtado et al. (their Fig. 8) as well as by Thmopson et al. [Ref: Hundred-fold enhancement in far-field radiative heat transfer over the blackbody limit. Nature 561, 216-221, 2018]. As Fernández-Hurtado et al. pointed out, the TE guided modes have the shorter propagation length compared to the TM-asymmetric modes, so the TE guided modes contribute more to the directional emissivity compared to the TM-asymmetric in their case. Here, we found the third one (TM-symmetric) has the largest confinement as well as the shortest propagation length (shorter than that of the TE). Therefore, the TM-symmetric could also contribute to the hemispherical emissivity of the thin films (while it might not be important in the cases of

Fernández-Hurtado et al. and *Thompson et al.* that focused *directional* thermal radiation between two objects, due to its very short propagation length). Therefore, both the TE and TM contribute to the hemispherical emissivity of the thin films in our specific case (namely, far field hemispherical emissivity from a single object), with a slightly larger TE contribution.

Now the main question is why is the nanoribbon hemispherical emissivity higher than that of thin film? We think there are probably two effects. First, the finite width provides a better coupling between the propagating light in the free space and the guided modes in the ribbons, similar to the grating effect shown by *Greffet et al.* [Ref: Coherent emission of light by thermal sources. *Nature* 416, 61-64, 2002]. Note that the wavelength of the guided modes are from 0.8 to 1.8 μm for the TM-symmetric modes and $\sim 10 \mu\text{m}$ (i.e., thermal wavelength) for the TM-asymmetric and TE modes, both of which match well with our designed nanoribbon width (~ 5 and $10 \mu\text{m}$). Second, the finite ribbon width can support higher order modes due to the resonance, especially for the TM-symmetric (see the dispersion relation for the nanoribbon as shown below, which we calculated in the revised version of the paper, Fig. 5). These higher order modes contribute to the higher density of state at the resonance frequency, and hence higher emissivity at this frequency.

Figure 5. Numerical mode analysis of nanoribbons. a, Dispersions of nanoribbons by numerical modelling. The numerical results (symbols) were compared to the analytical results of light line, single interface and an infinite slab. **b,** Zoomed-in plots of **a**. The insets represent the electric field

intensity distributions in the cross-section of a 5- μm wide nanoribbon at the four colour-filled symbols.

More specific response with supporting materials were described below, in accordance with the detailed comments from the reviewer.

There are many more issues that the authors should amend and address before the paper can become publishable. In what follows, I list them following the order in which they appear in the manuscript:

(Reviewer 2-Q1)1) Already in the abstract the authors talk about coherent thermal emission. This term is rather confusing because the coherence of the radiation is not analyzed or probed in these experiments. Actually, what the authors seem to mean is narrowband thermal emission (i.e. more monochromatic), as opposed to the broadband emission of a black body. I would personally change this because it is very misleading (it took me a while to understand what the authors meant by coherent).

Response: We are sorry to make confusion with the important term we introduced. The term of the coherent thermal emission has been used in the literature (e.g., Ref: J.-J. Greffet *et al.*, Coherent emission of light by thermal sources, *Nature*, 406, 61-64 (2002)) to describe the phenomena involving the light-matter interactions. Especially, when light energy allows to be shrunk with smaller wavelengths under the conditions to support surface polaritons, $\text{real}(\epsilon_{\text{SiO}_2}) < 0$, and guided modes, $\text{real}(\epsilon_{\text{SiO}_2}) < \epsilon_{\text{air}}$, absorption cross-section can be extensively enhanced over the geometric cross-sectional area. As the reviewer mentioned, the ultimate feature is shown as the narrow band thermal emission which is targeting to be more monochromatic, unlike the broadband emission of a black body. Similarly, the concept of the coherent thermal emission was introduced by J.-J. Greffet *et al.* The coherent thermal emission using a grating array of SiC was demonstrated by surface phonon polaritons. The related studies have been more extensively conducted with optical antenna structures using plasmonic materials in UV-Vis region [Ref: Ingvarsson *et al.*, Enhanced thermal emission from individual antenna-like nanoheaters, *Optical Express*, 15(18), 11249 (2007)], which is corresponding to a much higher energy regime than room temperature.

To avoid the confusion, we revised our *abstract* and *introduction* to more specifically define the feature of the “coherent thermal emission”, and also add more description

On page 1, “Coherent thermal emission deviates from the thermal emission predicted for a blackbody by the classical Planck’s law **with a narrow spectrum and strong directionality.**”

On page 3, added “**Recent work has shown the feasibility of spectral and spatial control of far field thermal emission from engineered microstructures. For instance, Greffet *et al.* demonstrated coherent emission, featured with a narrow spectral and angular range, achieved by the grating of a polaritonic material (i.e., SiC) with a grating period on the order of λ_T to couple to SPhPs.¹⁴ More recently, through fluctuational electrodynamics modelling, Fernández-Hurtado *et al.*¹⁵ showed highly directional emissivity from anisotropic polaritonic structures, which was found to originate from the guided modes whose dispersion relation becomes highly dispersive around the Reststrahlen band. This feature resulted in super-Planckian thermal radiation between two anisotropic objects when they are oriented along the dominant emission direction, which was also experimentally demonstrated by Thompson *et al.*¹⁶”**

On page 15, Added “**The geometry-dependent peak shift in the metallic regime indicates the coherent effect, as opposed to the monotonic change in the dielectric regime (near 12 μm wavelength).**”

(Reviewer 2-Q2) 2) When Fig. 1 is first discussed in the introduction, it is unclear how the emissivity is computed. This is a non-trivial issue since notice that the authors do not provide results for the total emissivity, which is actually the most important quantity in this work. This only becomes clear much later in the manuscript. Moreover, the meaning of the angle theta has to be somehow guessed since it is not indicated in that figure. Again, this only becomes clear later in the manuscript.

Response: We thank the reviewer’s careful examination. We agree to the comments that the partial information on the modeling from Fig. 1 would confuse readers at the beginning of the paper, although we intended to show the general idea of the selective spectral enhancement of the emissivity.

In order to make a better flow to describe our work, we follow the steps by: 1) introducing the developed measurement platform, 2) describing experimental observations, and 3) analyzing the results through combined analytical and numerical modeling in the main manuscript.

Following the reviewer's comments, we removed the modeling results of nanoribbons at the certain angle from Fig. 1d while adding more detailed discussion on angular dependent emittance in Figs. S12-S13 (for thin films) and S15-S16 (for nanoribbons) as shown below (see more detailed response for this figure in Reviewer 2-comment #6).

Figure S12. Plots of spectral and directional emissivity of TM mode of a 100 nm thin film.

Figure S13. Plots of spectral and directional emissivity of TE mode of a 100 nm thin film.

Figure S15. Plots of spectral absorption efficiency of nanoribbon ($W = 5 \mu\text{m}$) with various incident angles, where the polarized electric fields are on the planes including the length.

Figure S16. Plots of spectral absorption efficiency of nanoribbon ($W = 5 \mu\text{m}$) with various incident angles, where the polarized electric fields are on the plane normal to the length.

By doing so, we could clarify the directions of polarization and propagation before introducing the modeling results in Fig. 4c with the inset picture. Also, similarly, we start mentioning the results with different angles after we show the inset to indicate each angle, θ and φ in Fig. 4b.

Figure 4. Enhanced emissivity with anisotropic nanoribbons. **a**, Plots of the extracted emissivity at room temperature for ribbon widths of 6.28 and 11.5 μm . The grey bar represents the computed emissivity of an infinitely wide thin film of the same thickness as the ribbons, i.e., 100 nm. **b**, Plots of directional emissivity (ϵ_{dir}) of a nanoribbon with $W=5 \mu\text{m}$ at a wavelength of 9.5 μm , where the incoming plane wave has propagation (k) directions where the polarisation directions are normal to k in x - y plane and x - z plane, respectively for each $\phi=90^\circ$ and $\theta=90^\circ$ cases, and its incident angle is controlled by θ and ϕ as shown in the insets. **c-d**, Enhanced absorption cross-sectional area as a function of wavelength, where the absorption cross-sectional area (σ_{abs}) is normalised by the geometrical cross-section (σ_{geom}) for various widths (c) and thicknesses (d).

(Reviewer 2-Q3)3 In page 6, the authors discuss the results by first analyzing the SPhPs, see Eq. 1. As explained above, these are not the most relevant modes for the thermal emission of these nanoribbons. Neglecting the TE modes, the authors are ignoring a very important part of the underlying physics.

Moreover, Eq. 1 is not well explained in the text and it is not obvious whether it applies to the system under study. First, the authors should clearly explain that this is the dispersion relation of TM modes for an infinite slab. But, is it obvious that the infinite slab approximation is justified in this case? The width of the nanoribbons is not that large. Second, this expression is taken from a work where the SPhPs of a 2D material were studied. Is it obvious that this formula applies to a system with a thickness of 100 nm? The properties of those modes (and TE modes as well) can be described with standard theories of dielectric waveguides (see e.g. Ref. 34 for the case of an infinite slab).

Response: In response to the reviewer's comments, we did more detailed mode analysis, including for both TE and TM polarizations, from analytical modeling using an infinite slab as a simplified case, to numerical modeling to consider the 3D nanoribbon structures. As the reviewer pointed out the significant contributions of TE modes, we summarized the equations for analytical models and compared the different influences of each modes. We can get three different dispersion at each thickness, namely symmetric and asymmetric TM polarization, and one mode with TE polarization.

The detailed derivation for the analytical results of dispersion relations is now included in the Supplementary Note S8 and the results are shown in Figs. S19 and S21 below. In addition, to address the reviewer's question about the validity of the Eq.1 in the paper, we replaced it to a more general analytical model to calculate the dispersion and showed that **the results are the same as the ones obtained from the simplified Eq.1, as shown in Fig. S19a for thin films**. We later used numeric model to obtain the dispersions for nanoribbons (with finite width).

Note that the algebraically simplified equations (original Eq. 1) only count the zeroth order mode. The dispersion by Eq. 1 considers the Fabry-Perot resonance, and the equation can be shown as [Ref: Dai et al., Tunable Phonon Polaritons in Atomically Thin van der Waals Crystals of Boron Nitride, Science, 343 (6175), 1125 (2014)]:

$$k_p = q(\omega) + i\kappa(\omega) = \frac{i}{t} \left[2 \arctan \left(i \frac{\epsilon_1}{\epsilon_2} \right) + \pi l \right]$$

where l is an integer for the higher order modes, and the dispersion curves of fundamental modes were shown in Fig. S19 with $l=0$. Solid lines represent the results from the equations we summarized above, and the symbols from the reference equation. The difference from the reference study with h-BN and our study with SiO₂ is the applied permittivity. 2D materials such as h-BN require the consideration of the anisotropic permittivity with different directions, but SiO₂ possesses isotropic permittivity.

Figure S19. Dispersion relations of thin films with various thicknesses, in the cases of symmetric (a) and asymmetric (b) configurations of electric fields. (c,d) Plots of propagating length, $1/(2\kappa)$, of symmetric (c) and asymmetric (d) modes, respectively. TM polarization was considered as described in Fig. S18.

Figure S21. (a) Dispersion relations and (b) propagating length of thin films with various thickness, in the case of TE polarized wave-guided modes.

During this review process, we recognized that using the simplified equation from Ref. 27 (now Ref. 29) can confuse readers although it results in the same dispersion plot for thin films. Therefore, we **changed the Eq. 1 in the manuscript to the general form with more detailed procedures in the supporting information.** Please note that the results from the general equation shown in revised Eq (1) and the simplified equation from Ref. 27 (now Ref. 29) are the same.

On page 7, “We determined the dispersion of an infinite slab using the following analytical formula:

$$\frac{\varepsilon_{\text{SiO}_2}}{\varepsilon_0} = -\frac{k_{t,\text{SiO}_2}}{k_{t,0}} \coth\left(\frac{t}{2i} k_{t,\text{SiO}_2}\right), \quad (1)$$

where ε_0 and $\varepsilon_{\text{SiO}_2} (= \varepsilon' + i\varepsilon'')$ are the permittivity in vacuum and SiO_2 ,²⁸ respectively; $k_{t,i}$ is the transverse vector ($k_{t,i}^2 + k_p^2 = \varepsilon_i k_0^2$) in the medium i ; k_p is the propagating vector along the surface ($k_p = q + i\kappa$, where q and κ are the real and imaginary part of the momentum vector, respectively); and k_0 is the free space vector. Eq. (1) is the solution for the even modes of two branches of the surface waves by electric fields at the top and bottom surfaces, and this case represents the dominant contributions for the efficient absorption by confining the energy from the free space into the small physical cross-section area.²⁹ The confinement becomes more significant with a smaller thickness, due to the stronger interaction of the two symmetric evanescent surface waves

(Fig. 1c). The analytical equations are summarized in the Supplementary Information (see note #8) for the major available fundamental modes by different polarizations (namely, symmetric TM, asymmetric TM, and TE modes).

Furthermore, we compared the dispersions with TE and TM polarizations using the equations from Ref. 34 (now Ref. 15):

$$k_{TE} = \frac{\omega}{c} \sqrt{1 + \left(\frac{\omega t (\epsilon_{SiO_2} - 1)}{2c} \right)^2}$$

$$k_{TM} = \frac{\omega}{c} \sqrt{1 + \left(\frac{\omega t (\epsilon_{SiO_2} - 1)}{2c \epsilon_{SiO_2}} \right)^2}$$

The overlapped dispersion curves are shown in Fig. R1. The solid lines are based on the equations from the reference for both TE and TM, and the dotted lines are the results from Figs. S19 and S21 to be directly compared. Please note that for the TM mode comparison, we adopted the result for the asymmetric TM configuration (even branches of two surface waves). In *Fernández-Hurtado et al.* (Ref. 34 in the original manuscript), it was found that the TE mode has a larger attenuation constant (or shorter propagation length) compared to the TM-asymmetric modes, and hence could contribute significantly to the directional emissivity. In our case where the hemispherical emissivity was measured (and also calculated for the thin films), we also found that the TM-symmetric modes have very short propagation length (even shorter than that of the TE modes) as well as strong confinement effect [Ref: C. J. Fu et al., Planar heterogeneous structures for coherent emission of radiation, *Optics Letters*, 30 (14), p1873, 2005], so we believe this would make these modes contribute to the emissivity significantly as well.

Fig. R1. Plots of dispersions calculated using the equations from Ref. 34, for (a-b) TM polarization and (c-d) TE polarization.

In addition, following the reviewer's excellent suggestion of studying the effect of the finite width, we extended this mode analysis study from the infinite slab to the nanoribbon with the finite width and thickness. In the end, we observed very close dispersion to the thin film results with both 5 and 10 μm widths (see Fig. 5a) due to the smaller effective SPhP wavelength ($\sim 1 \mu\text{m}$) along the surface for the thin structure as we could expect from the analytical results of the thin slab. However, the nanoribbons also show higher order modes due to the resonance effect (Fig. 5). Therefore, we think there are two effects contributing to the enhanced hemispherical emissivity in ribbons compared to the thin film limit, as we discussed earlier: (i) the finite width provides a better coupling between the propagating light in the free space and the guided modes in the ribbons,

similar to the grating effect shown by Greffet *et al.* Note that the wavelength of the guided modes (0.8 to 1.8 μm for the TM-symmetric modes and $\sim 10 \mu\text{m}$ for the TM-asymmetric and TE modes) match well with our designed nanoribbon width (~ 5 and $10 \mu\text{m}$). (ii) the finite ribbon width can support higher order modes due to the resonance, leading to the higher density of state and hence higher emissivity at the resonance frequency.

Figure 5. Numerical mode analysis of nanoribbons. a, Dispersions of nanoribbons by numerical modelling. The numerical results (symbols) were compared to the analytical results of light line, single interface and an infinite slab. **b**, Zoomed-in plots of **a**. The insets represent the electric field intensity distributions in the cross-section of a $5\text{-}\mu\text{m}$ wide nanoribbon at the four colour-filled symbols.

(Reviewer 2-Q4) 4) The term “thermal fin model” is not so well-known. The authors should provide a reference or refer to the Methods section.

Response: We thank the reviewer for the suggestion. We now more prominently describe it in the *Methods* section with general and also detailed information on the fin equation.

On page 19, “A fin has high aspect ratios in dimensions to extend the surface area. Therefore, the temperature change along the length has the major contribution while it can be assumed that the temperature across the smaller dimension is constant. For this reason, the fin

structure can possess the limited heat conduction and more effectively radiate heat, thus it has been widely used to cooling devices as a heat sink⁴⁴.

The temperature profile can be described as Fig. S2. By defining the temperature profile along the x-axis, parallel to the length, above the ambient temperature (θ) as below.

Figure S2. Schematics of a SiO₂ ribbon with radiative heat loss ($m > 0$), where θ is a temperature rise.

$$q = C_1 \cosh(mx) + C_2 \sinh(mx), \text{ where } C_1 \text{ and } C_2 \text{ are unknown constants.} \quad (7)$$

$$q = T - T_o \quad (8)$$

The general solution introduces the feature of exponential temperature drop while propagating along the x-axis. The decaying rate is weighted by m , so-called fin parameter, and it is determined by:

$$m = \sqrt{\frac{hP}{kA_c}} \quad (9)$$

$$h @ 4\epsilon\sigma T^3 \quad (10)$$

where h is the heat transfer coefficient, ϵ is the emissivity, σ is the Stefan–Boltzmann constant, and T is the temperature. Quantifying the m parameter enables to obtain the ϵ .

To specify the unknown parameters from the solution for θ , here we summarize the boundary conditions for temperatures at $x=0$ and $x=L$:

$$\theta|_{x=0} = C_1 = q_h \quad (11)$$

$$\theta|_{x=L} = q_h \cosh(mL) + C_2 \sinh(mL) = q_s \quad (12)$$

$$C_1 = q_h \quad (13)$$

$$C_2 = \frac{q_s - q_h \cosh(mL)}{\sinh(mL)} \quad (14)$$

where θ_h and θ_s are the temperature rise at the ends of the fin. By integrating two ends to the suspended metal beams, θ_h and θ_s are measured in this thermometry platform. Therefore, θ can be represented as:

$$\theta = q_h \cosh(mx) + \frac{q_s - q_h \cosh(mL)}{\sinh(mL)} \sinh(mx) \quad (15)$$

Next, we summarize the boundary conditions for heat flux at $x=0$ and $x=L$.

For heat flux at $x=0$ (heating side),

$$Q_h = -kA_c \left. \frac{\partial \theta}{\partial x} \right|_{x=0} = -kA_c m C_2 \quad (16)$$

For heat flux at $x=L$ (sensing side),

$$Q_s = -kA_c \left. \frac{\partial \theta}{\partial x} \right|_{x=L} = -kA_c \{mC_1 \sinh(mL) + mC_2 \cosh(mL)\} \quad (17)$$

To meet overall energy balance,

$$Q_s = G_b q_s \quad (18)$$

$$-kA_c \{mC_1 \sinh(mL) + mC_2 \cosh(mL)\} = G_b q_s \quad (19)$$

$$-kA_c m \left(q_h \sinh(mL) + \frac{q_s - q_h \cosh(mL)}{\sinh(mL)} \cosh(mL) \right) = G_b q_s \quad (20)$$

$$q_h \sinh^2(mL) - q_h \cosh^2(mL) + q_s \left\{ \cosh(mL) + \frac{G_b}{kA_c m} \sinh(mL) \right\} = 0 \quad (21)$$

$$-q_h + q_s \left\{ \cosh(mL) + \frac{G_b}{kA_c m} \sinh(mL) \right\} = 0 \quad (22)$$

$$q_s = \frac{q_h}{\cosh(mL) + \frac{G_b}{kA_c m} \sinh(mL)} \quad (23)$$

$$\left. \frac{\theta_s}{\theta_h} \right|_{m>0} = \frac{1}{\cosh(mL) + \frac{G_b}{kA_c m} \sinh(mL)} \equiv \gamma. \quad (24)$$

As shown in Fig. S2c, the temperature profile become linear in the absence of heat loss ($h = 0$, and eventually $m = 0$).

$$Q_s = \frac{Q_h}{1 + \frac{G_b}{kA_c} L} = \frac{Q_h}{1 + \frac{G_b}{G_s}} = \frac{G_s}{G_s + G_b} q_h \quad (25)$$

$$G_s (q_h - q_s) = G_b q_s \quad (26)$$

$$q_s = \frac{G_s q_h}{G_b + G_s} \quad (27)$$

$$\left. \frac{\theta_s}{\theta_h} \right|_{m=0} = \frac{G_s}{G_b + G_s} \cong \frac{G_s}{G_b} \equiv \gamma_{lossless}. \quad (28)$$

Finally, we can define the ratio factor, Δ as:

$$D \equiv \frac{g}{g_{lossless}} = \frac{G_a}{G_s} = \frac{k_a}{k_s}. \quad (29)$$

where G_a is the measured apparent thermal conductance. By comparing the expected ratios of γ to the ratios of the measured conductivity, ε can be extracted.”

(Reviewer 2-Q5)5 In page 10 the authors state that: “As the lengths of the ribbons were much longer than the penetration depth³⁴, we assumed a single ϵ value to fit all the samples with the same width but different lengths”. This is not quite correct. In Ref. 34 it is explained that depending on the frequency (in the range relevant for the thermal emission), the propagation length can be larger than 1 mm. Thus, this assumption has to be justified with an analysis of the corresponding modes or, ideally, with a theoretical analysis of the length-dependence of the emissivity.

Response: We apologize the wrong assumption and the consequent confusing statement, and thank the reviewer for her/his correction.

We first would like to clarify that a **single ϵ value for different lengths is a purely experimental result**. As shown in Figs. 3e-f, each of the fitting lines has a single fitting **parameter in ϵ** over all lengths. The plots were used to show the process to fit the experimental results using the fin model, including averaged, upper and lower limit of emissivity. The averaged and error values by assuming the constant emissivity were summarized in Fig. 6a.

Figure 3 e–f, Plots of temperature ratios, Δ , between the heating and sensing beams as a function of temperature. Δ is defined as $\gamma/\gamma_{lossless}$ and is a measure of the radiative heat loss. **The experimental results (symbols) are compared to the fitted fin model (dotted lines), and the coloured area indicates the uncertainties of the fitted emissivity values.**

Figure 6a. Plots of emissivity as a function of temperature. The inset shows the modelled emissivity of an infinite thin film.

As the reviewer correctly pointed out, the theoretical calculation of length dependence of emissivity is perhaps a complicated question, which we admit is currently lacking in our current study and needs further theoretical study. For more detailed mode analysis for the nanoribbon structures, we conducted numerical modeling using COMSOL Multiphysics, and added the results in Fig. 5. The mode analysis was calculated with the assumption of the infinite length, and the finite width and thickness. Therefore, we could compare the dispersions of nanoribbons (finite width) and thin film (infinite width).

Figure 5. Numerical mode analysis of nanoribbons. a, Dispersions of nanoribbons by numerical modelling. The numerical results (symbols) were compared to the analytical results of light line, single interface and an infinite slab. **b,** Zoomed-in plots of **a**. The insets represent the electric field intensity distributions at the cross-section of 5 μm width nanoribbons at the four colour-filled symbols.

As shown in Fig. 5, the dominant modes with high q are very close to the dispersion of the thin film. Therefore, similar to the analysis from the thin film, it can be concluded that symmetric configurations of electric fields at the surfaces dominantly support the energy confinement. In addition, we observed the electric field intensity distributions at the cross-section of nanoribbons. Indeed it showed high intensity along the width. The lowest order of modes, marked with the red circle, are well overlapped with the fundamental mode available from the thin film dispersion. As getting higher orders, we could clearly observe the coherent resonance along the width. Please note that still the propagating direction is along the length, and the diverse forms of polarization increase the number of modes. Although the higher order of modes lowers q , still multiple modes possess significantly high q , compared to the light line. Further, the shift in q is smaller with the narrower width.

Corresponding to the real part (q) of propagating vector ($k_p = q + i\kappa$), we also studied the imaginary counterpart (κ). The propagating length was calculated by $1/2\kappa$, and it is shown in Fig. 5c. As the reviewer pointed out, broad range of propagating length can be supported. Therefore,

to expect the contributions on emissivity and to study the length-dependent emissivity, more intensive modeling covering high aspect-ratios (e.g. the ratio of length to thickness is 1000s : 1) is required.

Figure 5c. Propagating lengths of nanoribbons with 5 and 10 μm width, corresponding to the dispersion in **a-b**.

We have calculated the mode analysis of an infinite slab and nanoribbons. Also, we modeled absorption-cross sections with the assumptions of infinite lengths. However, we believe that more extensive modeling with all finite and anisotropic dimensions is beyond the scope of this study. Here, we aim to introduce our newly developed measurement platform and to show the enhanced far-field emission of a single polar dielectric nanoribbon by surface phonon polaritons.

Therefore, we removed the statement that the length of nanoribbons is much longer than the penetration depth, and make it as an open question for the theoretical study (similarly, also see the response to Reviewer 2- comment #6), while we describe the feature as we observed from our experimental study.

On Page 12, “To extract ε , we directly compared the experimental and fitted Δ values. **We used a single ε value to fit all the samples with the same width but different lengths.** The dashed line in Fig. 3 represents the Δ value based on the averaged ε value showing the best fit over the entire length range, and the shaded area indicates the uncertainty of the fitting. **The uncertainty could originate from the slight variation in the actual widths of the fabricated samples with different**

lengths. Also, it may include the length-dependent behaviour with the nature of broad range of propagating length as we shall see later.”

On Page 15, “Similar to the analytical model using a thin film, we also conducted mode analysis study of nanoribbons with the finite width and thickness. We compared the propagating vector of the thin film to that of nanoribbons calculated using COMSOL. The ribbon structure assumes infinite length and the k_p is parallel to the length. The two finite dimensions in thickness and width resulted in multiple modes, as shown in Fig. 5. The dominant modes with high q are very close to the dispersion of the thin film. Therefore, similar to the analysis from the thin film, it can be concluded that symmetric configurations of electric fields at the surfaces dominantly support the energy confinement. In addition, we observed the electric field intensity distributions at the cross-section of nanoribbons. Indeed, it showed high intensity along the width. The lowest order of modes, marked with the red circle, are well overlapped with the fundamental mode available from the dispersion relation of a thin film. For higher order modes, we could clearly observe the coherent resonance along the width (Fig. 5b). Please note that still the propagating direction is along the length, and the diverse forms of polarization increase the number of modes. Although the higher order modes have lower q values, still multiple modes possess significantly higher q , compared to the light line. Further, the shift in q is smaller with the narrower width (Fig. 5b). Corresponding to the real part (q) of propagating vector ($k_p = q + i\kappa$), we also studied the imaginary counterpart (κ). The propagating length was then calculated by $1/2\kappa$, shown in Fig. 5c. Broad range of propagating length from 100s nm to ~1 mm, can be supported. It implies that the anisotropic structure could have the length-dependent emissivity. In our experiments (Figs 3e-f), the fitted lines assume a constant ε over all the lengths up to 600 μm . The error range covers the potential length-dependent behaviour. All modelling in this study has been done with an infinite length, but more rigorous theoretical study, such as fluctuational electrodynamics¹⁵ could be a potential interesting study for the non-traditional far-field radiation.”

(Reviewer 2-Q6)6) The most important result is the observation of the enhanced emissivities. This observation, in turn, shows the failure of the classical Planck’s law. But then, the obvious question is whether these results can be explained by fluctuational electrodynamics, which is the rigorous theory describing any thermal radiation problem, irrespective of the size of the objects involved.

This is not addressed at all in this work. The calculation of the total emissivity of silica nanoribbons has been provided in Ref. 34. Would it be possible to present similar results for the exact dimensions of these experiments? I understand that the nanoribbons are very long and their modelling may be challenging, but it is worth trying since this is, in my opinion, the most important question related to the main results of this work. If it is not possible, the authors should at least mention that this is an interesting open problem.

Response: We appreciate the comments to introduce the different approach to model emissivity by fluctuational electrodynamics, which has been nicely done in Ref. 34 (now Ref. 15). As the reviewer recognized, indeed, the most challenging part to model for this highly anisotropic structures is to have extra-fine mesh to cover the shrunk wavelength and thickness (100 nm) for very long ribbons (100s micron) . Based on the numerical mode analysis, the SPhP wavelength can be even down to ~500 nm, so the required mesh needs to be as small as ~50 nm. In addition, the small mesh needs to be extended through the entire length (up to 600 μm). Therefore, in this study, we rather used the modeling as a tool for mechanistic study, i.e., to analyze the optical modes available in the thin nanoribbon structures with an infinite length, and to show their distinguished feature in thin films and ribbons, relative to bulk SiO_2 , rather than aiming to calculate hemispherical emission. Indeed, the emissivity calculation is an interesting open question, and it can be a follow up study and potentially solved by the methodology developed in Ref. 34. We highly appreciate the reviewer's recognition of the challenge of directly modeling the nanoribbons using fluctuational electrodynamics and his/her openness of allowing us to leave it as an open question. We believe that with the methodology introduced in Ref. 34 (now Ref. 15), such calculations can be done in follow up studies either by us or others.

In response to the reviewer's comment, we address the introduced modeling method by fluctuational electrodynamics in Ref. 34 (now Ref. 15), but we leave the rigorous theoretical study as an open question, which in fact is our follow-up study. This point is directly connected to the response to Reviewer 2-Q5.

On Page 12, "To extract ϵ , we directly compared the experimental and fitted Δ values. **We used a single ϵ value to fit all the samples with the same width but different lengths.** The dashed line in

Fig. 3 represents the Δ value based on the averaged ε value showing the best fit over the entire length range, and the shaded area indicates the uncertainty of the fitting. The uncertainty could originate from the slight variation in the actual widths of the fabricated samples with different lengths. Also, it may include the length-dependent behaviour with the nature of broad range of propagating length as we shall see later.”

On Page 13, “This enhancement can be understood from the strong resonant nature of the antenna-like nanoribbons, which results in an effective absorption cross-section (σ_{abs}) that is significantly higher than the geometrical one (σ_{geom}).¹⁵ Fernández-Hurtado *et al.*¹⁵ suggested theoretical guidelines to overcome the far-field limit by Planck’s law and have systematically studied the far field radiation heat transfer from and between various sub-wavelength dielectric objects through the theoretical and numeric modelling. The computational study introduced the enhanced far-field thermal conductance between two anisotropic objects by increasing absorption efficiency, which was subsequently verified experimentally by Thompson *et al.*¹⁶ Similarly, we modelled σ_{abs} of the nanoribbons using a finite element method (FEM) approach in COMSOL.”

On Page 15, “Similar to the analytical model using a thin film, we also conducted mode analysis study of nanoribbons with the finite width and thickness. We compared the propagating vector of the thin film to that of nanoribbons calculated using COMSOL. The ribbon structure assumes infinite length and the k_p is parallel to the length. The two finite dimensions in thickness and width resulted in multiple modes, as shown in Fig. 5. The dominant modes with high q are very close to the dispersion of the thin film. Therefore, similar to the analysis from the thin film, it can be concluded that symmetric configurations of electric fields at the surfaces dominantly support the energy confinement. In addition, we observed the electric field intensity distributions at the cross-section of nanoribbons. Indeed, it showed high intensity along the width. The lowest order of modes, marked with the red circle, are well overlapped with the fundamental mode available from the dispersion relation of a thin film. For higher order modes, we could clearly observe the coherent resonance along the width (Fig. 5b). Please note that still the propagating direction is along the length, and the diverse forms of polarization increase the number of modes. Although the higher order modes have lower q values, still multiple modes possess significantly higher q , compared to the light line. Further, the shift in q is smaller with the narrower width (Fig. 5b). Corresponding to

the real part (q) of propagating vector ($k_p = q + i\kappa$), we also studied the imaginary counterpart (κ). The propagating length was then calculated by $1/2\kappa$, shown in Fig. 5c. Broad range of propagating length from 100s nm to ~ 1 mm, can be supported. It implies that the anisotropic structure could have the length-dependent emissivity. In our experiments (Figs 3e-f), the fitted lines assume a constant ε over all the lengths up to 600 μm . The error range covers the potential length-dependent behaviour. All modelling in this study has been done with an infinite length, but more rigorous theoretical study, such as fluctuational electrodynamics¹⁵ could be a potential interesting study for the non-traditional far-field radiation.”

(Reviewer 2-Q7)7) The main argument used to explain the main observation is basically borrowed from Ref. 34. In this sense, I think that it is fair to refer to this work in a more explicitly way. Otherwise, one may give the impression that this explanation is proposed for the first time in this work.

Response: We thank the reviewer for the suggestions. We fully agree that Fernández-Hurtado *et al.* (Ref. 34 in the original manuscript) provided important insight to show the enhanced absorption efficiency and far-field emissivity. Also, indeed, there have been very few studies on nontraditional far-field radiation, compared to the near-field counterparts, and Ref. 34 is the first such study we are aware of. Thus, we revised our introduction to more directly and prominently mention the previous work (including Ref. 34, now Ref. 15) and to connect it more directly to this study. Thus, we included more descriptions as below:

On page 3, line 2, added “Recent work has shown the feasibility of spectral and spatial control of far field thermal emission from engineered microstructures. For instance, Greffet *et al.* demonstrated coherent emission, featured with a narrow spectral and angular range, achieved by the grating of a polaritonic material (i.e., SiC) with a grating period on the order of λ_T to couple to SPhPs.¹⁴ More recently, through fluctuational electrodynamics modelling, Fernández-Hurtado *et al.*¹⁵ showed highly directional emissivity from anisotropic polaritonic structures, which was found to originate from the guided modes whose dispersion relation becomes highly dispersive around the Reststrahlen band. This feature resulted in super-Planckian thermal radiation between two anisotropic objects when they are oriented along the dominant emission direction, which was also experimentally demonstrated by Thompson *et al.*¹⁶

These earlier findings provide important insights into the design of an emitter with coherent far field thermal emission.”

On Page 13, “This enhancement can be understood from the strong resonant nature of the antenna-like nanoribbons, which results in an effective absorption cross-section (σ_{abs}) that is significantly higher than the geometrical one (σ_{geom}).¹⁵ Fernández-Hurtado *et al.*¹⁵ suggested theoretical guidelines to overcome the far-field limit by Planck’s law and have systematically studied the far field radiation heat transfer from and between various sub-wavelength dielectric objects through the theoretical and numeric modelling. The computational study introduced the enhanced far-field thermal conductance between two anisotropic objects by increasing absorption efficiency, which was subsequently verified experimentally by Thompson *et al.*¹⁶ Similarly, we modelled σ_{abs} of the nanoribbons using a finite element method (FEM) approach in COMSOL”

On page 17, line 7, added “This was also clearly shown by Fernández-Hurtado *et al.*.”

(Reviewer 2-Q8)8) When the authors discuss the direction dependence of the absorption cross section, they actually show results for the TE polarization (see Fig. 4c). These results clearly demonstrate that the TE modes play a crucial role and, in turn, contradict the central conclusion that the SPhPs dominate the physics. In other words, those results should have led the authors to conclude that their explanation is not consistent.

Response: We thank the reviewer for the thorough examination and we again apologize for the confusion caused by us. As mentioned in our initial response to the reviewer and in our revised manuscript, we fully agree that the TE modes contribute significantly to the emissivity of thin films and ribbons. This can be seen in our modeled results for hemispherical emissivity of thin films (in supplementary note S7 and Fig. S14). In fact, in our specific case studied here (hemispherical emissivity of 100 nm thick SiO₂ nanoribbons), we think both TE and TM modes contribute significantly. In reality, because we handle diverse angles and also the ribbons have four facets (unlike thin films), ϕ and θ , each plane has **combined influences of both the TE and TM polarization**. In other words, we cannot easily define TE or TM for *nanoribbons* and thus our nomenclature in the original manuscript has been revised in this revision. Our modeling was done with the fixed incident TE or TM polarized wave while rotating the ribbons by different angles. Unlike the infinite film which has only one finite dimension in thickness, the nanoribbon

structures have two finite dimensions in both width and thickness. Therefore, although we have an incident wave with the TE or TM polarization for one specific facet, the other perpendicular planes will meet a different polarization, relatively defined by φ and θ .

For instance, here we draw an explicit case where it showed the opposite polarizations with the rotated structures. Figure S17a shows incident TE wave on the x-y plane, which is normal to the length. On the plane of incidence, the TE wave does not have an electric field component which is polarized normal to the surface (width or thickness) on x-y plane. On the other hand, as the nanoribbons are 3D structures, we can move our view point to one of perpendicular planes to see how different faces are affected by the TE incident on x-y plane. Figure S17b shows the example on the z-y plane. In this normal plane to x-y plane, there is an electric field component polarizing across the surface. Therefore, one incident wave can excite both TE and TM modes in the 3D structures.

Figure S17. Relative polarization modes at different facets.

We understand it would make readers confused without showing the more absolute definition. Thus, we clarified the directions of electric field polarization and propagation by indicating directly on the x, y or z-axis arrows for each specific case, rather than using the term of TE and TM polarization for these 3D structures with rotations. It will provide more absolute information of the polarization directions. Accordingly, we revised Fig. 4b and Figs. S15-S16.

Figure 4b. Plots of directional emissivity (ϵ_{dir}) of a nanoribbon with $W= 5 \mu\text{m}$ at a wavelength of $9.5 \mu\text{m}$, where the incoming plane wave has propagation (k) directions where the polarisation directions are normal to k in x - y plane and x - z plane, respectively for each $\phi= 90^\circ$ and $\theta= 90^\circ$ cases, and its incident angle is controlled by θ and ϕ as shown in the insets.

Figure S15. Plots of spectral absorption efficiency of nanoribbon ($W = 5 \mu\text{m}$) with various incident angles, where the polarized electric fields are on the planes including the length.

Figure S16. Plots of spectral absorption efficiency of nanoribbon ($W = 5 \mu\text{m}$) with various incident angles, where the polarized electric fields are on the plane normal to the length.

Reviewer #3

In this work, Shin et al. demonstrated an experimental platform to quantify the thermal radiation from individual nanoribbons. Using this platform, they quantified the size dependence of the emissivity of nanoribbons. The authors designed the experiment well, and the contribution of the radiative heat transfer is extracted in a well-controlled way. The emissivity is obtained from the contribution of radiation. The authors were intended to show that ribbons that can support thermally excited localized phonon polaritonic resonances can create coherent thermal emission and result in an emissivity that is larger than the thin-film counterpart or ribbons that do not support localized resonances. Although I like the technique and the platform of this work, I think the analysis of the phonon polaritons in the ribbon and the connection with the measured emissivity are not clear, and therefore, the intended demonstrations were not convincing. I cannot recommend its acceptance in its current form.

Response: We thank the reviewer for his/her positive evaluations on our experimental platform, and for the careful examinations and suggestion to strengthen our argument. We did more detailed mode analysis to clarify the dominant mechanisms to enhance the emissivity. More specific and detailed explanations are described with the following point-by-point responses.

(Reviewer 3-Q1)1. Equation (1) is quoted from Ref. 27, which is for a thin hBN film. hBN is a hyperbolic material, and its waveguide dispersion may not be directly applicable for SiO₂ ribbons. For SiO₂ in its Reststrahlen band, the authors may want to treat it as a thin metallic film and consider the coupled surface phonon polariton, which has a different dispersion from Eq. (1) in this work. The wavelength results shown in Fig. 1c, therefore, cannot be trusted. Also, what is the value of l ? Based on Eq. (1), shouldn't there be more than one order of modes?.

Response: We thank the reviewer for pointing out the validity of the Eq. 1. We resolve the issue by applying a more general equation to achieve dispersions of surface phonon polaritons in the manuscript. We show that the results from the more general equation and from the original equation (1) would be identical.

The Eq.1 quoted from Ref. 27 (now in Ref. 29) is the simplified form of symmetric surface waves with thin films. The equation was summarized based on hBN but the treatment can be easily adjusted with permittivity vectors. As we considered the isotropic permittivity in the SiO₂ volume, unlike the anisotropic permittivity in hBN, the equation can also be correctly applied. In addition, within the Reststrahlen band, the real part of permittivity of SiO₂ becomes negative, which can be considered as metallic. Therefore, the results can represent the dispersion of SiO₂, similar to that of hBN.

However, we recognized that the simplified equation, rather than the general form, would confuse the readers without detailed explanations. Thus, we added more detailed procedures to calculate the dispersions of surface phonon polaritons in a thin film in the Supporting Information (Figs. S19 and S21, also shown below). We also introduced the equation in the manuscript using the general form. The original equation (1) includes the higher order of modes with $l>0$ in case Fabry-Perot resonances can be supported with small dissipation. To directly compare the results to the general analytical equations, we considered $l=0$ for the fundamental mode. Later we also show that this mode has the major contribution in the next comment below (Reviewer 3- Comment #2). Furthermore, the two cases of dispersions were calculated with symmetric and asymmetric

configurations of the two surface waves at the top and bottom surfaces of a thin film. Eventually, we confirmed the identical results with different sources of equations as in Fig. S19a, so the wavelength calculation shown in Fig. 1c would not change.

Figure S19a. Dispersion relations of thin films with various thickness, in the case of symmetric configurations of electric fields.

On page 7, “We determined the dispersion of an infinite slab using the following analytical formula (see Supplementary note #S8):

$$\frac{\varepsilon_{SiO_2}}{\varepsilon_0} = -\frac{k_{t,SiO_2}}{k_{t,0}} \coth\left(\frac{t}{2i} k_{t,SiO_2}\right), \quad (1)$$

where ε_0 and $\varepsilon_{SiO_2} (= \varepsilon' + i\varepsilon'')$ are the permittivity in vacuum and SiO_2 ,²⁸ respectively; $k_{t,i}$ is the transverse vector ($k_{t,i}^2 + k_p^2 = \varepsilon_i k_0^2$) in the medium i ; k_p is the propagating vector along the surface ($k_p = q + i\kappa$, where q and κ are the real and imaginary part of the momentum vector, respectively); and k_0 is the free space vector. Eq. (1) is the solution for the even modes of two branches of the surface waves by electric fields at the top and bottom surfaces, and this case represents the dominant contributions for the efficient absorption by confining the energy from the free space into the small physical cross-section area.²⁹ The confinement becomes more significant with a

smaller thickness, due to the stronger interaction of the two symmetric evanescent surface waves (Fig. 1c). The analytical equations are summarized in the Supplementary Information (see note #8) for the major available fundamental modes by different polarizations (namely, symmetric TM, asymmetric TM, and TE modes).

In the Supporting Information, we added the detailed procedures to get dispersions:

Figure S18. Schematics of (a) a thin film structure consisting of a medium 2 (SiO_2) surrounded by medium 1 (air), and (b) symmetric and (c) asymmetric configurations of TM polarized surface waves.

Electromagnetic interactions between two adjunct surfaces can be significant with the poles of the reflective coefficient, r , determined by the Fresnel equation,

$$r_{TM} = \frac{\varepsilon_2 k_{t,1} - \varepsilon_1 k_{t,2}}{\varepsilon_2 k_{t,1} + \varepsilon_1 k_{t,2}} \quad (\text{S1})$$

where ε_i is the complex permittivity, $k_{t,i}$ is the transverse vector ($k_{t,i}^2 + k_p^2 = \varepsilon_i k_0^2$) in the medium i , k_p is the propagating vector along the surface ($k_p = q + i\kappa$), where q and κ are the real and imaginary part of the momentum vector, respectively), and k_0 is the free space vector. To figure out the fundamental modes with two surfaces, we solve

$$1 - r^2 e^{2ik_{t,2}t} = 0. \quad (\text{S2})$$

The above equation yields solutions for the poles of the complex reflectivity. TM polarization can support surface phonon polaritons when the real part of permittivity is negative. The analytical solutions for the above equations can be achieved by considering two different configurations of surface waves at each one of the two interfaces. For the symmetric electric field configuration of TM polarization as shown in Fig. S18b,

$$r_{TM} = e^{ik_{t,2}t} = \frac{\varepsilon_2 k_{t,1} - \varepsilon_1 k_{t,2}}{\varepsilon_2 k_{t,1} + \varepsilon_1 k_{t,2}} \quad (\text{S3})$$

$$(\varepsilon_2 k_{t,1} + \varepsilon_1 k_{t,2}) e^{ik_{t,2}t} = \varepsilon_2 k_{t,1} - \varepsilon_1 k_{t,2} \quad (\text{S4})$$

$$\varepsilon_2 k_{t,1} (1 - e^{ik_{t,2}t}) = \varepsilon_1 k_{t,2} (1 + e^{ik_{t,2}t}) \quad (\text{S5})$$

$$\frac{\varepsilon_2}{\varepsilon_1} = -\frac{k_{t,2}}{k_{t,1}} \coth\left(\frac{t}{2i} k_{t,2}\right). \quad (\text{S6})$$

In the thin film regime ($k_{t,2}t \ll 2$), the equation can be simplified in the form of:

$$\frac{\varepsilon_2}{\varepsilon_1} = -\frac{2i}{t} \frac{1}{k_{t,1}}. \quad (\text{S7})$$

where $k_{t,1}$ is positive. In the case where the highly confined guided modes ($k_{t,i} = \sqrt{\varepsilon_i k_0^2 - k_p^2} \approx ik_p$) are supported, the further simplified form follows:

$$k_p = -\frac{2}{t} \frac{\varepsilon_1}{\varepsilon_2}. \quad (\text{S8})$$

Note that this algebraically simplified equation only counts the zeroth order mode. With the consideration of the Fabry-Perot resonance, the equation can be shown as²⁸:

$$k_p = q(\omega) + i\kappa(\omega) = \frac{i}{t} \left[2 \arctan\left(i \frac{\varepsilon_1}{\varepsilon_2}\right) + \pi l \right] \quad (\text{S9})$$

where l is an integer for the higher order modes, and the dispersion curves of fundamental modes were shown in Fig. S19 with $l=0$.

Similarly, for the asymmetric electric field configuration for TM polarization as shown in Fig. S18c,

$$r_{TM} = -e^{ik_{t,2}t} = -\frac{\varepsilon_2 k_{t,1} - \varepsilon_1 k_{t,2}}{\varepsilon_2 k_{t,1} + \varepsilon_1 k_{t,2}} \quad (\text{S10})$$

$$\frac{\varepsilon_2}{\varepsilon_1} = -\frac{k_{t,2}}{k_{t,1}} \tanh\left(\frac{t}{2i} k_{t,2}\right). \quad (\text{S11})$$

In the thin film regime ($k_{t,2}t \ll 2$), the aforementioned equation can be simplified in the forms of:

$$\frac{\varepsilon_2}{\varepsilon_1} = -\frac{t}{2i} \frac{k_{t,2}^2}{k_{t,1}} \quad (\text{S12})$$

where $k_{t,1}$ is positive.

By comparing the simplified equations from symmetric and asymmetric dispersions, it is clear that two distinct fundamental modes have opposite thickness-dependent behaviors. While the symmetric mode possesses more bounded modes as the film thickness decreases, the asymmetric mode becomes closer to the light line with decreasing film thickness, as shown in Figs. S19a-b. The thickness-dependent behaviors are clearly shown in the Reststrahlen band ranging from 200 to 220 Trad/s , and the features become deviating from the light line with the dominant confinement effect at the surface. Therefore, we believe it is plausible that surface phonon polaritons are supported. The shrunk wavelength along the surface, unlike the free-space wavelength, efficiently enhances the absorption cross-section. Correspondingly, the higher confinement (high q) yields shorter propagating length, which implies more efficient absorption.

Figure S19. Dispersion relations of thin films with various thickness, in the cases of symmetric (a) and asymmetric (b) configurations of electric fields. (c,d) Plots of propagating length, $1/(2\kappa)$ of symmetric and asymmetric modes, respectively. TM polarization was considered as described in Fig. S18.

Figure S20. Schematics of (a) a thin film structure consisting of a medium 2 surrounded by 1, and (b) TE wave guide mode.

TE polarization does not have an electric field component normal to the surface as shown in Fig. S20, thus it cannot support surface waves unless permeability, μ has negative values to form polaritons by magnetic fields in an infinite slab while it still makes reflection at the interface as:

$$r_{TE} = \frac{\mu_2 k_{t,1} - \mu_1 k_{t,2}}{\mu_2 k_{t,1} + \mu_1 k_{t,2}}, \text{ where } \mu_1 = \mu_2 = 1. \quad (\text{S13})$$

The only available solution to satisfy positive $k_{t,1}$ is following:

$$r_{TE} = -e^{ik_{t,2}t} = \frac{k_{t,1} - k_{t,2}}{k_{t,1} + k_{t,2}} \quad (\text{S14})$$

$$1 = -\frac{k_{t,2}}{k_{t,1}} \tanh\left(\frac{t}{2i} k_{t,2}\right) \quad (\text{S15})$$

$$1 = -\frac{t}{2i} \frac{k_{t,2}^2}{k_{t,1}}, \text{ in the thin film limit } (k_{t,2}t \ll 2). \quad (\text{S16})$$

The analytically calculated dispersions were shown in Fig. S16(a). There is a sharp change in the propagating vector, q at around 200 Trad/s, deviating from the light line. More specifically right below ω_{LO} (200 Trad/s), the real part of ϵ_{SiO_2} becomes smaller than that of air ($\epsilon_{air}=1$) down to 199 Trad/s. In this regime, the repeated internal reflections within the SiO_2 can generate the guided modes in the SiO_2 . However, the effective energy range is very narrow to meet $\epsilon_{SiO_2} < \epsilon_{air}$, furthermore, the thin structure diminishes the guiding effect. The lowest order of the guide modes requires a half of the wavelength at least in dimensions. In addition to the guiding effect, the SiO_2 also can be considered as a lossy dielectric layer with high refractive index, especially at around 190 Trad/s because of the peaks of both real and imaginary parts of permittivity. Therefore, the

effective refractive index of a thin film surrounded by air will be closer to that of air with smaller lossy volume.

Figure S21. (a) Dispersion relations and (b) propagating length of thin films with various thickness, in the case of TE polarized wave-guided modes. ”

Based on the results using the revised Eq. (1), we revised the Fig. 1 as:

Figure 1. Anisotropic nanoribbon for localised resonance by polaritons. **a**, Schematic illustration of a long nanoribbon with rectangular cross section with thickness (t) and width (W). Here, t is thinner than the skin depth (δ), leading to significant transmitted intensity of the electromagnetic wave perpendicular to the ribbon ($|E|_T$) compared with the absorbed intensity ($|E|_\varepsilon$). Otherwise, the optical response in the parallel direction to the plane would yield higher emission because the cross-sectional length scale with the integer multiples of half wavelength supports the localised resonance modes, enhancing the absorption cross-section. **b**, Plots of the real and imaginary permittivity of SiO₂. The region of the Reststrahlen band is coloured. **c**, Dispersion relation of SPhP supported by thin films of different thickness, calculated from the approximate solution (Eq. 1). The energy range is within the Reststrahlen band. **d**, **SPhP wavelength and propagating length for a 100 nm thick thin film, corresponding to the dispersion curve.**

(Reviewer 3-Q2) 2. The authors may want to provide information on the propagation length of the surface phonon polaritons. This is important to understand the size dependence of the resonance. If the propagation length is too smaller, then the ribbon width would not matter.

Response: We appreciate the reviewer's suggestion to include the propagating length. As stated in the response to the previous comment (Reviewer 3-comment #1), we supplemented analytical modeling results for both the real and imaginary parts of propagating vectors in the Supplementary Information. From the imaginary part of the propagating vector, we obtained the propagation length ($=1/2\kappa$). The results for the major contributing mode (e.g. symmetric TM polarization) were summarized in Figs. 1c-d. Also, we added numerical modeling results of nanoribbons in the main manuscript (Fig. 5).

In this response, we want to clarify that we studied mode analysis of nanoribbon with the propagating direction *along the length*. Both width and thickness determine the finite cross-section. Similarly, in the 2D model for an infinite slab (thin film), thickness is the finite dimension. Therefore, here we discuss the length-dependent behavior compared to the propagating length, rather than the width-dependent property. In the next response (Reviewer 3-comment #3), we show the width- and thickness-dependent behavior by comparing the dimensions with the effective SPhP wavelength.

Firstly, we added the propagating length plot in Fig. 1d as a result of the analytical models for thin films, as shown in the previous response #1.

Figure 1. **c**, Dispersion relation of SPhP supported by thin films of different thickness, calculated from the approximate solution (Eq. 1). The energy range is within the Reststrahlen band. **d**, SPhP wavelength and propagating length for a 100 nm thick thin film, corresponding to the dispersion curves.

To calculate the propagating length for the nanoribbons, which do not have analytical solutions, we conducted numerical modeling using COMSOL Multiphysics, and added the results in Fig. 5 of the main manuscript. The mode analysis was calculated with the assumption of an infinite length, and a finite width ($W=10$ and $5 \mu\text{m}$, respectively) and thickness ($t=100 \text{ nm}$), similar to our experimental conditions. The influence of the finite width will be discussed in the next response (Reviewer 3-comment #3) for geometry-dependent resonant behaviors using the real part of propagating vectors (Figs. 5a-b). To model the propagating length, we achieved the imaginary counterpart (κ), corresponding to the real part (q) of propagating vector ($k_p = q + i\kappa$). The propagating length was calculated by $1/2\kappa$, and shown in Fig. 5c.

Figure 5. Numerical mode analysis of nanoribbons. **a**, Dispersions of nanoribbons by numerical modelling. The numerical results (symbols) were compared to the analytical results of light line, single interface and an infinite slab. **b**, Zoomed-in plots of **a**. The insets represent the electric field intensity distributions in the cross-section of 5- μm width nanoribbons at the four colour-filled symbols. **c**, Propagating lengths of nanoribbons with 5 and 10 μm widths, corresponding to the dispersion in **a-b**.

From these results, we could observe the broad-range of propagating length. For the specific modes to support surface phonon polaritons with symmetric configurations, short mean free path, $\sim 100\text{s nm}$ can be available for efficient absorption along the length. In the other modes, the propagation lengths are longer, up to $\sim 500 \mu\text{m}$. It implies that along with the higher energy

confinement in the Reststrahlen band, the electromagnetic waves can be efficiently absorbed. As much as we maximize the absorption efficiency within this energy regime (with high Q factor), higher emission at the specific direction and spectral energy can be achieved. This feature is the novelty of coherent far-field emission, compared to the traditional heat transfer, which only allows isotropic and incandescent thermal radiation.

In our experiments, we used constant emissivity over the lengths from 50 to 600 μm . During this revision process, we realized that broad-range of propagating length makes it difficult to assume. Therefore, we revised our manuscript by mentioning that the error bar in the measured emissivity includes the potential emissivity variation with different lengths.

On Page 12, “To extract ε , we directly compared the experimental and fitted Δ values. We used a single ε value to fit all the samples with the same width but different lengths. The dashed line in Fig. 3 represents the Δ value based on the averaged ε value showing the best fit over the entire length range, and the shaded area indicates the uncertainty of the fitting. The uncertainty could originate from the slight variation in the actual widths of the fabricated samples with different lengths. Also, it may include the length-dependent behaviour with the nature of broad range of propagating length as we shall see later.”

On Page 15, “Corresponding to the real part (q) of propagating vector ($k_p = q + i\kappa$), we also studied the imaginary counterpart (κ). The propagating length was then calculated by $1/2\kappa$, shown in Fig. 5c. Broad range of propagating length from 100s nm to ~ 1 mm, can be supported. It implies that the anisotropic structure could have the length-dependent emissivity. In our experiments (Figs 3e-f), the fitted lines assume a constant ε over all the lengths up to 600 μm . The error range covers the potential length-dependent behaviour. All modelling in this study has been done with an infinite length, but more rigorous theoretical study, such as fluctuational electrodynamics¹⁵ could be a potential interesting study for the non-traditional far-field radiation.”

(Reviewer 3-Q3)3. The author may need to provide a model to describe the geometric dependence of the ribbon resonances. After all, this work is aimed to demonstrate coherent thermal emission. Without such a model, it is hard to connect the total radiation loss in a broad wavelength and angular range with coherent thermal emission.

Response: We thank the reviewer for the suggestion. To show the geometric-dependent absorption efficiency within the Reststrahlen band where surface phonon polaritons are supported, we calculated the enhanced absorption cross-section with various width and thickness values, as shown in Figs. 4c and 4d, respectively.

Figure 4. Enhanced emissivity with anisotropic nanoribbons. c-d, Enhanced absorption cross-sectional area as a function of wavelength, where the absorption cross-sectional area (σ_{abs}) is normalised by the geometrical cross-section (σ_{geom}) for various widths (c) and thicknesses (d).

As shown in Fig. 4c, the various finite width showed different peak intensities and locations within the Reststrahlen band. The 5 and 10 μm widths possess the peaks at ω_{LO} , which is the edge of the Reststrahlen band with the peak of the imaginary part of the permittivity of SiO_2 . As we shall see later in Fig. 5, 5 and 10 μm are relatively large, compared to the effective wavelength. Therefore, the enhancement can expand within the entire spectral range where the surface phonon polaritons are supported. In addition, they have similar absolute absorption cross section, thus it yields higher normalized value by the smaller geometric cross-sectional area in Fig. 4c. As we further reduce the width, the normalized absorption cross-section keep increasing. However, when the width is smaller than about the half of the effective wavelength of SPhP, it can no longer support the surface wave. Therefore, reducing the width down to 100 nm showed peak shifting to the shorter wavelength.

Similarly, we also conducted absorption efficiency calculation with different thicknesses with a constant width of 5 μm (Fig. 4d). The thicker ribbon would result in less interactions between the two evanescent waves at the top and bottom surfaces. Therefore, it showed smaller enhancement.

We made the following changes in the manuscript.

On Page 13, “We now seek to understand the enhancement of the coherent emission in the nanoribbons, which was over 8.5 fold for the 6.28- μm -wide ribbon at 150 K compared with the thin-film limit calculated above. This enhancement is remarkable, considering that the spectral window for emission ($\omega_{TO} \sim 1251 \text{ cm}^{-1}$ to $\omega_{LO} \sim 1065 \text{ cm}^{-1}$ or from 8 to 9.5 μm) is now much narrower compared with the broad thermal spectrum. This enhancement can be understood from the strong resonant nature of the antenna-like nanoribbons, which results in an effective absorption cross-section (σ_{abs}) that is significantly higher than the geometrical one (σ_{geom}).³⁴ Fernández-Hurtado *et al.*¹⁵ suggested theoretical guidelines to overcome the far-field limit by Planck’s law and have systematically studied the far field radiation heat transfer from and between various sub-wavelength dielectric objects through the theoretical and numeric modelling. The computational study introduced the enhanced far-field thermal conductance between two anisotropic objects by increasing absorption efficiency, which was subsequently verified experimentally by Thompson *et al.*¹⁶ Similarly, we modelled σ_{abs} of the nanoribbons using a finite element method (FEM) approach in COMSOL.”

On Page 14, “Here, two plane waves propagating along the length with normal polarisation were simultaneously exposed to evaluate their averaged contribution, as shown in the inset in Fig. 4c. The modelling results of σ_{abs} , as observed in Fig. 4c, indeed showed strong enhancement within the Reststrahlen band (Fig. 4c). The results are clearly distinguished from the lossy dielectric regime at around 12 μm wavelength. The peak enhancement factors ranged from 25 to 55 in ribbons of different widths. The enhancement increased when the ribbon width was reduced from 10 μm to 500 nm. From our results on the dispersion relations of SPhPs in a 100-nm-thick SiO_2 film (Fig. 1c), the wavelengths of the SPhPs ($2\pi/q$) ranged from 0.8 to 1.8 μm at around 9 μm . Therefore, the ribbons with widths down to 500 nm, which is larger than half of the shortest SPhP wavelength, can still support localised resonance SPhP modes.

Our modelling results also indicate that further reduction of the ribbon width below 500 nm would rather lead to smaller enhancement. For example, the peak enhancement factor for 100 nm width is 45, which is smaller than that of the 500 nm width (~55), see Fig. 4c. This difference occurs because when the width is reduced to less than half the wavelength of certain modes of SPhPs, the localised resonance modes can no longer be supported within the width of the ribbons. Therefore, as the width decreases, a greater portion of the localised SPhP modes become unavailable. This result highlights the importance of the anisotropic nanoribbon design with suitable widths in maximising the resonance effect and resultant coherent thermal emission, as an isotropic cross section such as that in a cylindrical rod would be unable to support certain resonance modes and consequently lead to a much smaller emissivity (e.g., $\varepsilon = 0.025$ in a 100-nm-diameter nanorod¹⁸). The geometry-dependent peak shift within the metallic regime (8 to 9.5 μm) indicates the coherent effect, as opposed to the monotonic change in the dielectric regime (near 12 μm wavelength).

We also modelled the enhanced σ_{abs} with different thicknesses. The thicker ribbon would result in the less overlaps of two evanescent waves at the top and bottom surfaces. Therefore, it yields smaller enhancement as shown in Fig. 4d. The enhanced σ_{abs} is closely correlated with the enhancement of the measured ε . Therefore, we can attribute the largely enhanced ε from the nanoribbons to the coherent emission within the Reststrahlen band caused by the strong SPhP resonance.”

For more detailed mode analysis for the nanoribbon structures, we conducted numerical modelling using COMSOL Multiphysics, and added the results in Fig. 5. The mode analysis was calculated with the assumption of an infinite length, and a finite width and thickness. Therefore, we could compare the dispersions of nanoribbons (finite width) and thin film (infinite width).

Figure 5. Numerical mode analysis of nanoribbons. **a**, Dispersions of nanoribbons by numerical modelling. The numerical results (symbols) were compared to the analytical results of light line, single interface and an infinite slab. **b**, Zoomed-in plots of **a**. The insets represent the electric field intensity distributions at the cross-section of $5 \mu\text{m}$ width nanoribbons at the four colour-filled symbols. **c**, Propagating lengths of nanoribbons with 5 and $10 \mu\text{m}$ widths, corresponding to the dispersion in **a-b**.

As shown in Fig. 5, the dominant modes with high q are very close to the dispersion of the thin film. Therefore, similar to the analysis from the thin film, it can be concluded that symmetric configurations of electric fields at the surfaces dominantly support the energy confinement. In addition, we observed the electric field intensity distributions at the cross-section of nanoribbons.

Indeed, it showed high intensity along the surface consisting of width and thickness. The lowest order of modes, marked with the red circle, are well overlapped with the fundamental mode available from the thin film dispersion. At higher orders, we could clearly observe the coherent resonance along the width (inset in Fig. 5b). Please note that still the propagating direction is along the length, and the diverse forms of polarization increase the number of modes. Although the higher order of modes lowers q , still multiple modes possess significantly high q , compared to the light line. Further, the shift in q is smaller with the narrower width. Similarly, we can expect a lower number of modes for if the width is further reduced to below the wavelength of the surface waves. It is consistent with the results shown in Fig. 4: ribbons of 500 nm and 100 nm widths have limited support of SPhP, especially for the longer wavelength range.

In addition, we also calculated dispersions for different thickness. As we could expected from the analytical mode analysis (Fig. 1), we observed a less pronounced confinement effect with increasing thickness, as shown in Fig. S23.

Figure S23. Dispersions of nanoribbons by numerical modelling with (a) 100 nm and (b) 500 nm thickness.

On Page 15, “Similar to the analytical model using a thin film, we also conducted mode analysis study of nanoribbons with the finite width and thickness. We compared the propagating vector of the thin film to that of nanoribbons calculated using COMSOL. The ribbon structure assumes infinite length and the k_p is parallel to the length. The two finite dimensions in thickness and width

resulted in multiple modes, as shown in Fig. 5. The dominant modes with high q are very close to the dispersion of the thin film. Therefore, similar to the analysis from the thin film, it can be concluded that symmetric configurations of electric fields at the surfaces dominantly support the energy confinement. In addition, we observed the electric field intensity distributions at the cross-section of nanoribbons. Indeed, it showed high intensity along the width. The lowest order of modes, marked with the red circle, are well overlapped with the fundamental mode available from the dispersion relation of a thin film. For higher order modes, we could clearly observe the coherent resonance along the width (Fig. 5b).

(Reviewer 3-Q4)4. The author may want to double check the unit used for intensity in Fig. 1d.

Response: We thank the reviewer for the correction. As the reviewer pointed out, the unit needs to be changed to $\text{W}/\text{m}^2\text{-sr}\text{-}\mu\text{m}$ from $\text{W}/\text{m}^2\text{-}\mu\text{m}$ to consider the fact that the spectral intensity is based on the spectral and directional emittance.

During the revision, we considered the comment from reviewer 2 (Reviewer 2-Q2) that the information for the modelling result presented in Fig. 1d is limited because it is at the beginning of the paper, and it only becomes clearer in the later part of the manuscript. Therefore, we removed the initial spectral and directional results from Fig. 1, so as to organize the flow of the results and figures in a better sequence. More modelling results are now added in Figures 4 and 5, as we have discussed earlier. Figure 1 was revised as below.

Figure 1. Anisotropic nanoribbon for localised resonance by polaritons. **a**, Schematic illustration of a long nanoribbon with rectangular cross section with thickness (t) and width (W). Here, t is thinner than the skin depth (δ), leading to significant transmitted intensity of the electromagnetic wave perpendicular to the ribbon ($|E|_T$) compared with the absorbed intensity ($|E|_E$). Otherwise, the optical response in the parallel direction to the plane would yield higher emission because the cross-sectional length scale with the integer multiples of half wavelength supports the localised resonance modes, enhancing the absorption cross-section. **b**, Plots of the real and imaginary permittivity of SiO₂. The region of the Reststrahlen band is coloured. **c**, Dispersion relation of SPhP supported by thin films of different thickness, calculated from the approximate solution (Eq. 1). The energy range is within the Reststrahlen band. **d**, SPhP wavelength and propagating length for a 100-nm thick thin film, corresponding to the dispersion curve.

(Reviewer 3-Q5)5. The ribbon is not a Super-Planckian emitter, as Ref. 34 has proved. Therefore, it is not entirely obvious that why a ribbon should have a high hemispherical emissivity, which is an integrated effect, if it can support a localized resonance only at a given wavelength and a given direction. The authors may need to prove it at least numerically.

Response: As the reviewer commented, the ribbon is not a Super-Planckian emitter. We would like to clarify that we did not claim a Super-Planckian emitter in the paper, and our result is consistent with Ref. 34 (now Ref. 15). What we showed is that we observed emissivity of ~ 0.2 from the nanoribbons at room temperature (and higher values at lower temperatures). The emissivity is higher than that of a thin film (emissivity 0.062), but it is still lower than the bulk value of SiO₂ (~ 0.9). We attributed this enhancement (over the thin film limit) to the localized resonance of SPhP within the narrower spectral and directional range, which we think is the novel feature of the nanoribbon design. The small thickness (100 nm) gives us an inherently low broadband emissivity due to the lack of absorbing thickness (smaller than the skin depth of ~ 10 μm infrared, corresponding to 300 K), so the observed higher emissivity of the ribbons is mainly contributed by the localized resonance, which has the narrow spectral and directional distribution we are aiming for. As the reviewer mentioned, the hemispherical emissivity is an integrated effect over the entire spectral and directional ranges. That is why our measured emissivity (~ 0.2 at room T) is still much smaller than the enhancement factor of the absorption cross-section within the spectral and directional range we calculated (Fig. 4). Nevertheless, the fact that the emissivity is higher than the thin film limit underscores the prominent effect of the localized modes. Since the resonant phenomena can be supported within the narrow band region (Reststrahlen band), our it is difficult to go beyond the upper limit of the blackbody unless the enhanced absorption cross-section becomes extremely significant to dominate the integrated emissivity over the spectral and spatial distributions.

The spectral enhancement in the absorption cross-section we observed from the modelling is up to ~ 50 times over the geometrical one. But it would not be effective in leading to super-Planckian hemispherical emissivity, as the measured hemispherical emissivity is averaged over the spectral and angular distribution. This is the most distinct difference from the study by Thomson et al. [Ref: Hundred-fold enhancement in far-field radiative heat transfer over the blackbody limit]. The reported study observed ‘hundred-fold’ enhancement over the blackbody limit (i.e., super-Planckian) with directional radiation. By locating two objects in parallel with a far-field gap, directional emissivity could be measured. On the other hand, in our study, measuring total radiation loss would not show the enhancement as high as the spectral and directional enhancement. We believe two different experimental approaches would boost more diverse new experimental studies on far-field radiation.

From the reviewer's comment, we noticed the importance to clearly mention that our ribbons are not super-Planckian emitters and our results underscore the important role of the localized resonances to the observed higher emissivity (over the thin film limit). Thus, we added the following discussions in the manuscript as:

On Page 16, "These findings clearly demonstrate the merit of the anisotropic designs to selectively emphasise the coherent modes. Our observation for the enhanced emissivity over the thin film limit originates from the fact that the selective enhancement (in the spatial and spectral domains) outweighs the reduced absorption by the smaller volume, compared to the skin-depth (determined by $\lambda/4\pi k$, where k is the imaginary part of the complex refractive index). This point is different from the previous study by Thompson *et al.*¹⁶ As we observed from the angle-dependent absorption-cross section, the most significant enhancement occurred along the edge with sharper angles. It implies the strong directionality. Here we want to clarify that the observed enhancement does not mean the super-Planckian emission. The spectral enhancement we observed from the modeling is up to 55, but it is within a certain spectral and directional range and thus is still insufficient to result in a super-Planckian hemispherical emissivity, which is an integrated effect over the entire spectral and angular ranges. This was also clearly shown by Fernández-Hurtado *et al.*¹⁵ Nevertheless, the fact that the emissivity is higher than the thin film limit underscores the prominent effect of the localized modes.

"

We thank again for all the reviewers' time and efforts that have gone into the careful examination of our manuscript and for giving valuable and insightful comments. These comments have certainly helped us improve our manuscript in the revised form. We hope the revisions we made in the manuscript and our responses have addressed the review comments, and the manuscript in its revised form is considered suitable for publication in *Nature Communications*.

Reviewers' Comments:

Reviewer #1:

Remarks to the Author:

The authors have addressed my concerns and performed additional measurements to support their claim.

I stress my appreciation of the additional experimental work that have been performed I recommend publication in nature communications

Reviewer #2:

Remarks to the Author:

First of all, let me point out a technical problem in the revised manuscript. In the pdf version that I downloaded from the journal website there are multiple typos in the equations, which seem to originate from a Word-to-pdf conversion. The problems are mainly related to Greek letters, which often appear as Latin ones in the pdf file. This is certainly annoying and only with the help of the previous version I was able to follow and understand those equations.

Having said that, I would like to start by acknowledging that the authors have done a remarkable job addressing the criticisms and comments of the referees in the previous round. To begin with, and following the advice of the reviewer 1, in the revised version they present a series of control experiments that clearly show the reliability of their experimental technique. With respect to my comments and suggestions, I had mainly criticized the theoretical analysis presented in the previous version. In particular, I pointed out that the authors were making a serious mistake by ignoring the contribution of TE guided modes, which are expected to play a key role in these type of dielectric nanoribbons. Following my suggestions, the authors present now a largely improved analysis of the origin of the enhanced emissivity in the nanoribbons where, indeed, the TE modes are shown to play a critical role. Moreover, they make now a better connection with the theory results of Ref. 15, which are very relevant for the present work. I still miss a comparison of the emissivity values measured in this work with rigorous calculations done in the framework of fluctuational electrodynamics, but I agree with the authors that the focus of this paper is on the novel experimental technique and results and that comparison will hopefully be provided in a future publication. In any case, and as I explained in my previous report, the authors present here one of the few measurements on the thermal emission of subwavelength objects, which is one of the main open problems in the field of thermal radiation. Thus, and in view of the improved discussions in the revised version, I am now happy to strongly recommend the publication of this manuscript in Nature Communications without further revision.

Reviewer #3:

Remarks to the Author:

I appreciate the authors' long response. However, I still cannot recommend its acceptance. The explanation of the ribbon mode is not satisfying and I'm not sure about how to interpret the results in Fig. 5. I recommend the authors study some literature of graphene ribbons and see how they got the dispersion relationship of the coherent ribbon modes. This could help the authors to explain the peaks in the emissivity. Without such a formula, it is difficult to highlight the coherence nature of the thermal emission.

As a side point, the authors' argument that the original equation (1) is equivalent to the new equation. I am not sure I can agree with that. The hBN waveguide model is different from the coupled SPhP model. The fundamental order gives the same results does not mean they are equivalent. Also, the authors' may also remove the derivation of the dispersion of the coupled SPhP model since it is very standard and can be found in textbooks.

We thank all reviewers for their positive evaluation on our revised manuscript, and for reviewers 1 and 2 for their strong recommendation of the publication of our revised manuscript in *Nature Communications*. Here we respond to the remaining concerns of reviewer #3.

Reviewer #3:

I appreciate the authors' long response. However, I still cannot recommend its acceptance. The explanation of the ribbon mode is not satisfying and I'm not sure about how to interpret the results in Fig. 5. I recommend the authors study some literature of graphene ribbons and see how they got the dispersion relationship of the coherent ribbon modes. This could help the authors to explain the peaks in the emissivity. Without such a formula, it is difficult to highlight the coherence nature of the thermal emission.

As a side point, the authors' argument that the original equation (1) is equivalent to the new equation. I am not sure I can agree with that. The hBN waveguide model is different from the coupled SPhP model. The fundamental order gives the same results does not mean they are equivalent. Also, the authors' may also remove the derivation of the dispersion of the coupled SPhP model since it is very standard and can be found in textbooks.

Response:

Regarding the results in Fig. 5, we plot dispersion relations of SPhPs for SiO₂ thin films using both analytical solutions and COMSOL numeric calculations, and they showed the same results (Supplementary Fig. 22). We then numerically computed the dispersion relations and resonant modes in SiO₂ nanoribbons (Figs. 5a-b). Further, we obtained the propagation length in Fig. 5c which is determined by the imaginary part of effective refractive index. We think the results are valid. The fundamental and higher-order modes shown in Fig. 5b are also consistent with one's physical intuition. Furthermore, we want to emphasize that we focus on the SPhP modes in the IR regime from the polar dielectric SiO₂. This would be different from surface plasma polariton from electrons in graphene ribbons.

Second, our previous (approximate) equation (1) is indeed from the study of surface phonon polaritons in hBN [Ref: “Tunable Phonon Polaritons in Atomically Thin van der Waals Crystals of Boron Nitride”, *Science*, **343**, 1125 (2014)]. However, we want to emphasize again that even though the original equation (1) was used for hBN in the Science paper, it is generally applicable to describe SPhP dispersion relations of other polar dielectrics, including SiO₂ and SiC. In any case, to avoid any confusion, in the revised manuscript, instead of the approximate equation, we now used the new (full) equation (1) for all the dispersion relation calculations of thin films (shown in Fig. 5a), which are from textbooks and have been used for SPhPs in polar dielectric materials in numerous papers. Furthermore, our analytical results based on equation (1) for the thin films agree well with the exact numerical results from COMSOL modeling (Supplementary Fig. 22), and our results are also in agreement with those of Fernández-Hurtado *et al.* on SiO₂ films (Ref. 15).

We agree with the reviewer that the derivation can also be found in textbooks (which again shows that the formula is generally applicable to all kinds of polar dielectrics, not just hBN). Nevertheless, we believe that it is good to show the different energy dispersions associated with the exact geometries of our nanoribbons to highlight our anisotropic design. It would help readers better understand the three available modes, e.g., TE, symmetric-TM, and asymmetric-TM, and each mode has different dispersion as well as different geometric dependency. As we cannot describe all the details in the main manuscript, we prefer to still show the fundamental three modes in the Supplementary Information.

We highly appreciate the reviewer’s valuable suggestion during both rounds of reviews, which has helped us improved our manuscript.